# An oncopeptide regulates m$^6$A recognition by the m$^6$A reader IGF2BP1 and tumorigenesis

Song Zhu[1,2], Ji-Zhong Wang[1,2], De Chen[1,2], Yu-Tian He[1], Nan Meng[1], Min Chen[1], Rui-Xun Lu[1], Xin-Hui Chen[1], Xiao-Lan Zhang[1] & Guang-Rong Yan [1✉]

N$^6$-methyladenosine (m$^6$A) is the most prevalent modification in eukaryotic RNAs. The biological importance of m$^6$A relies on m$^6$A readers, which control mRNA fate and function. However, it remains unexplored whether additional regulatory subunits of m$^6$A readers are involved in the m$^6$A recognition on RNAs. Here we discover that the long noncoding RNA (lncRNA) *LINC00266-1* encodes a 71-amino acid peptide. The peptide mainly interacts with the RNA-binding proteins, including the m$^6$A reader IGF2BP1, and is thus named "RNA-binding regulatory peptide" (RBRP). RBRP binds to IGF2BP1 and strengthens m$^6$A recognition by IGF2BP1 on RNAs, such as *c-Myc* mRNA, to increase the mRNA stability and expression of *c-Myc*, thereby promoting tumorigenesis. Cancer patients with RBRP$^{high}$ have a poor prognosis. Thus, the oncopeptide RBRP encoded by *LINC00266-1* is a regulatory subunit of m$^6$A readers and strengthens m$^6$A recognition on the target RNAs by the m$^6$A reader to exert its oncogenic functions.

[1] Biomedicine Research Center, Guangzhou Municipal and Guangdong Provincial Key Laboratory of Protein Modification and Degradation, State Key Laboratory of Respiratory Disease, The Third Affiliated Hospital of Guangzhou Medical University, Guangzhou 510150, China. [2] These authors contributed equally: Song Zhu, Ji-Zhong Wang, De Chen. ✉email: jxygr007@yahoo.com

N⁶-methyladenosine (m⁶A) is the most prevalent modification of eukaryotic RNAs. RNA m⁶A methylation is reversibly and dynamically regulated by the RNA methyltransferases (i.e., the m⁶A writers) METTL3 and METTL14, and demethylases (i.e., the m⁶A erasers) FTO and ALKBH5[1–3]. The biological importance of RNA m⁶A modification relies on m⁶A-binding proteins (i.e., the m⁶A readers), which directly guide distinct biological functions[1,2]. A group of YT521-B homology (YTH) domain-containing proteins, including YTHDF1, YTHDF2, YTHDF3, YTHDC1, and YTHDC2, the IGF-2 mRNA-binding proteins, including IGF2BP1, IGF2BP2, and IGF2BP3, and the heterogeneous nuclear ribonucleoproteins hnRNPA2B1 and hnRNPC have been identified as m⁶A RNA readers that control mRNA fate by regulating mRNA stability, translation, splicing, structure, decay, and subcellular localization[4–8]. Recently, a handful of additional regulatory subunits of m⁶A methyltransferases, including WTAP, VIRMA, ZC3H13, HAKAI, and RBM15/15B, are found to play an important role in the installation of m⁶A on RNAs[1,9–12]. However, the regulatory subunits of m⁶A RNA readers and m⁶A demethylases remain unexplored.

Long noncoding RNAs (lncRNAs) comprise the largest proportion of mammalian noncoding transcripts. LncRNAs are defined as transcripts of more than 200 nucleotides that do not contain protein-coding open reading frames (ORFs). The ENCODE project demonstrates that the human genome contains more than 28,000 distinct lncRNA transcripts[13]. LncRNAs have been identified as significant factors in cells by regulating gene transcription (in *cis* or *trans*), chromatin remodeling, and mRNA splicing; producing endogenous small interfering RNAs (siRNAs) and microRNA precursors; changing protein localization; regulating protein activity; and organizing protein components[14,15]. The dysregulation of lncRNAs has been linked to the hallmarks of cancers and cancer patient survival time[16–18]. However, whether lncRNAs are involved in regulating the installation, removal or recognition of m⁶A on RNAs remains unexplored.

In this study, we discover that *LINC00266-1*, which was previously annotated as an lncRNA in *Homo sapiens* and whose functions are unknown, actually encodes a 71-amino acid (aa) peptide. The peptide mainly interacts with RNA-binding proteins, including the m⁶A reader IGF2BP1. Thus, we term the peptide "RNA-binding regulatory peptide" (RBRP). RBRP binds to the m⁶A reader IGF2BP1 and promotes the m⁶A recognition by IGF2BP1 on RNAs such as *c-Myc* mRNA to increase the *c-Myc* mRNA stability and level by enhancing the recruitment of the RNA stabilizers HuR, MATK3, and PABPC1. The RBRP oncopeptide, but not the lncRNA *LINC00266-1* itself, promotes colorectal cancer (CRC) tumorigenesis by enhancing m⁶A recognition-dependent *c-Myc* mRNA stability. The *LINC00266-1* lncRNA level and RBRP oncopeptide level are increased in cancer tissues and are highly metastatic cell sublines compared with their levels in the corresponding adjacent nontumor tissues and parent cell lines, respectively. Patients with CRC presenting with high RBRP oncopeptide levels exhibit more aggressive clinicopathological phenotypes and shorter survival times than patients presenting with low levels. Collectively, our findings reveal that an oncopeptide, RBRP, encoded by the uncharacterized lncRNA *LINC00266-1*, is an additional subunit of the m⁶A reader and regulates the m⁶A recognition by m⁶A readers on target RNAs and tumorigenesis.

## Results
### Identification of lncRNAs with coding potential
An lncRNA with coding potential must bind to the ribosome. The ribosome-bound RNAs in CRC SW480 cells and the corresponding metastatic SW620 cells were separately sequenced. The identified lncRNAs were examined in this study. Three hundred and eighty-four differentially expressed ribosome-bound lncRNAs with a fold-change ≥1.2 were identified between SW480 and SW620 cells (Supplementary Data 1). The coding potential of these 384 ribosome-bound lncRNAs was further analyzed using the ORF Finder program from the NCBI tools. Fifty-five lncRNAs with coding potential that were differentially expressed between SW480 and SW620 cells were discovered (Supplementary Data 1).

The predicted peptide/protein-coding ORFs of the ten lncRNAs with coding potential, namely *LINC00263*, *KTN1-AS1*, *ZNF205-AS1*, *LOC100126784*, *LINC00266-1*, *LINC00115*, *LOC286467 (FIRRE)*, *FLJ20021*, *LOC728752*, and *LOC284581*, were selected and individually fused to the N terminus of the start codon-mutated *GFP* vectors (GFPmut). Only the ORFs of the lncRNAs *LINC00266-1* and *FLJ20021* were able to be translated; the other eight lncRNAs did not have coding potential (Fig. 1a). The lncRNA *LINC00266-1* was further investigated here and *FLJ20021* will be investigated in other studies.

### The lncRNA *LINC00266-1* encodes an uncharacterized peptide, RBRP
The lncRNA *LINC00266-1* was previously annotated as an intergenic lncRNA (lincRNA) in *H. sapiens* (NR_040415) and its functions are unknown. We observed a short 213-nucleotide ORF with the potential to encode a 71-aa peptide (Supplementary Fig. 1). We termed this peptide RBRP. Sequence comparisons did not identify homologs of *LINC00266-1* or the RBRP peptide in any other species, and there were no matching proteins and known domains/motifs in RBRP, indicating that RBRP is an uncharacterized peptide.

We generated a series of Flag-fused constructs containing the ORF and the 5′-untranslated region (5′-UTR)-ORF of *LINC00266-1* and ATG-mutated start codon constructs to confirm that the start codon of the predicted ORF of the lncRNA *LINC00266-1* was active. Substantial expression of the RBRP-Flag fusion peptide was observed in *LINC00266-1* ORF-Flag (ORF)- and 5′UTR-ORF-Flag (5′U)-transfected cells, whereas the mutation of the start codon ATG (5′UTR-ORFmut-Flag, MT) abolished the translation of the predicted ORF in *LINC00266-1* (Fig. 1b, c).

We produced an antibody against the RBRP peptide to further detect and validate the RBRP peptide that was encoded by the *LINC00266-1* ORF in cells. The specificity of the anti-RBRP antibodies against the RBRP peptide was validated (Supplementary Fig. 2a, b). The RBRP-Flag fusion peptides were further validated using anti-RBRP antibodies in *LINC00266-1* ORF- and 5′UTR-ORF-transfected cells, whereas a mutation in the ATG start codon in the ORF of *LINC00266-1* abolished the detection of the RBRP fusion peptide (Fig. 1b, c). Collectively, the RBRP fusion peptide was translated into cells.

Finally, the *LINC00266-1* ORF-GFPmut fusion construct was transfected into HEK293T cells and the RBRP-GFP fusion peptide was immunoprecipitated using an anti-green fluorescent protein (GFP) antibody. Two unique peptide fragments in the RBRP peptide were identified using mass spectrometry (MS) (Fig. 1d). Collectively, these data indicate that *LINC00266-1*, which is previously annotated as a lncRNA, actually encodes a 71-aa peptide.

### The RBRP peptide is naturally, endogenously expressed
We detected the cellular RBRP peptide using anti-RBRP antibodies to validate the presence and expression of the natural, endogenous RBRP peptide in cells and tissues. Our developed anti-RBRP antibodies specifically detected the naturally endogenously

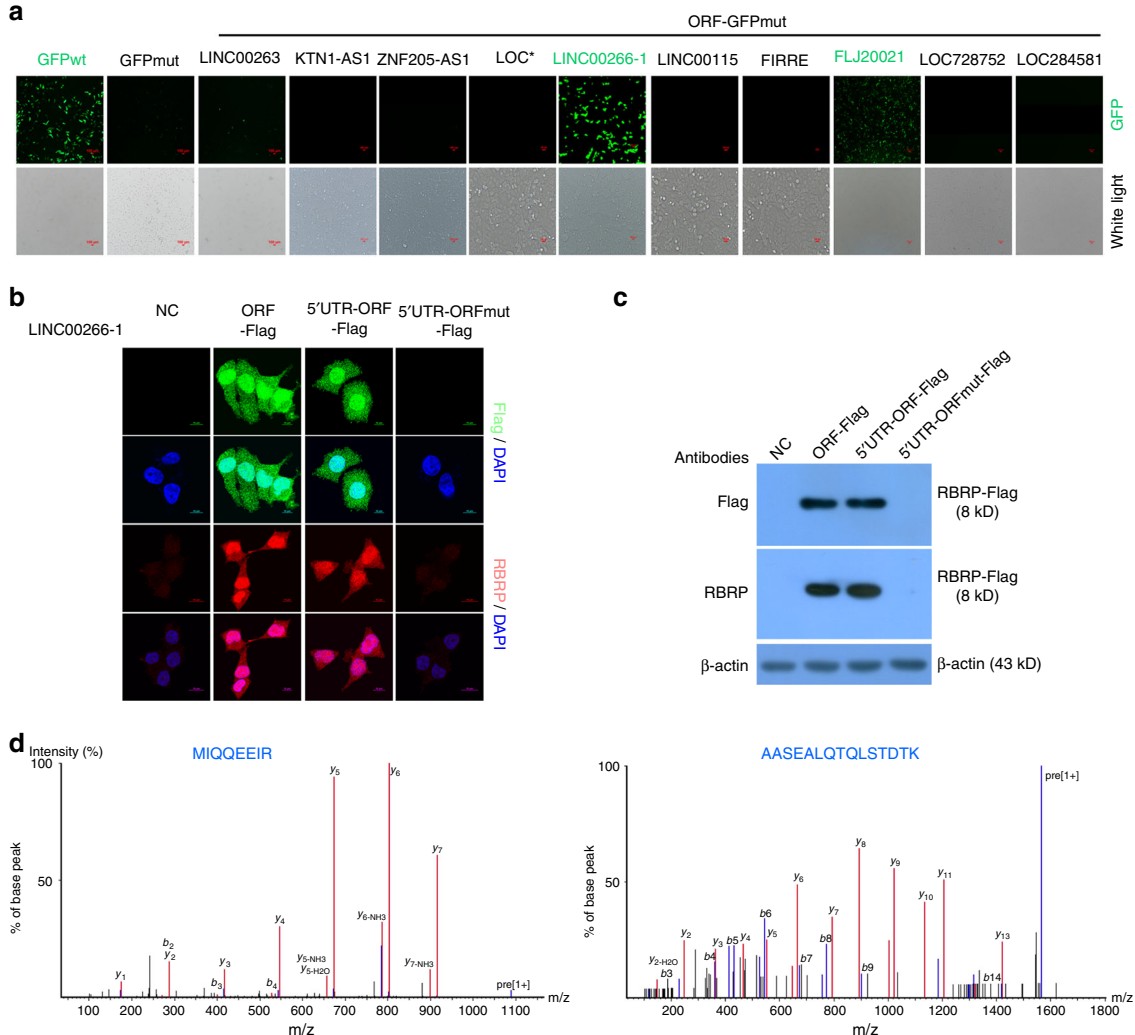

**Fig. 1 The lncRNA *LINC00266-1* encodes a 71-aa peptide, RBRP. a** The ORF-GFPmut constructs of ten indicated lncRNAs were transfected into HeLa cells and GFP fluorescence was detected. LOC* is *LOC100126784* and *FIRRE* is also named *LOC286467*. Scale bar: 100 µm. **b, c** The indicated constructs were transfected into HeLa cells for 48 h, immunostaining of the RBRP-Flag fusion peptide was detected with anti-Flag and anti-RBRP antibodies (**b**), and levels of the RBRP-Flag fusion peptide were detected by performing western blottings with anti-Flag and anti-RBRP antibodies (**c**). Scale bar: 10 µm. **d** Two unique peptides in the RBRP peptide were identified using mass spectrometry. Source data are provided as a Source Data file.

produced RBRP peptide (Supplementary Fig. 2c–e). The natural, endogenous RBRP peptide was validated in colorectal, breast, ovarian, and nasopharyngeal cancer cells (Fig. 2a, b). In addition, the RBRP peptide was confirmed to be naturally endogenously produced in five pairs of matched fresh primary CRC tissues and their corresponding adjacent nontumoral colorectal tissues (Fig. 2c). Finally, three unique peptide fragments in the natural, endogenous RBRP peptide were identified using MS analysis (Supplementary Fig. 3), further indicating that the natural, endogenous RBRP peptide was present in cancer tissues. Taken together, RBRP is naturally, endogenously, and widely expressed in various cells and tissues.

**Levels of the *LINC00266-1* lncRNA and RBRP peptide are increased in highly metastatic cancer cells and primary cancers.** According to our ribosome-bound RNA-sequencing data, *LINC00266-1* expression was upregulated in CRC SW620 cells compared with the expression in SW480 cells. We further confirmed that *LINC00266-1* lncRNA expression was upregulated in highly metastatic colorectal (SW620 and HCT-116[high]), ovarian

(SK-OV-3[high] and OVCAR-3[high]), nasopharyngeal (S18), and breast (MDA-MB-231[high]) cancer cell sublines compared with the expression in their parental cell lines (Supplementary Fig. 4a, b). The levels of the *LINC00266-1* lncRNA were also increased in five pairs of matched freshly frozen primary CRC tissues and their corresponding adjacent nontumoral colorectal tissues (Supplementary Fig. 4c, d). Unfortunately, no *LINC00266-1* expression data were obtained from the publicly accessible datasets The Cancer Genome Atlas and Oncomine.

Similar to *LINC00266-1* lncRNA levels, RBRP levels were also increased in highly metastatic colorectal, ovarian, nasopharyngeal, and breast cancer cell sublines compared with the levels in their parental cell lines, and the levels were also increased in the primary CRC tissues compared with the levels in the corresponding nontumoral colorectal tissues (Fig. 2b, c). Furthermore, an extensive tissue microarray analysis of 90 pairs of matched CRC tissues and corresponding nontumor samples of colorectal tissues was performed using immunohistochemistry (IHC) with anti-RBRP antibodies (Fig. 2d). Similar to the results in Fig. 2c, increased RBRP levels were observed in CRC tissues compared with the levels in adjacent nontumor samples of colorectal tissues

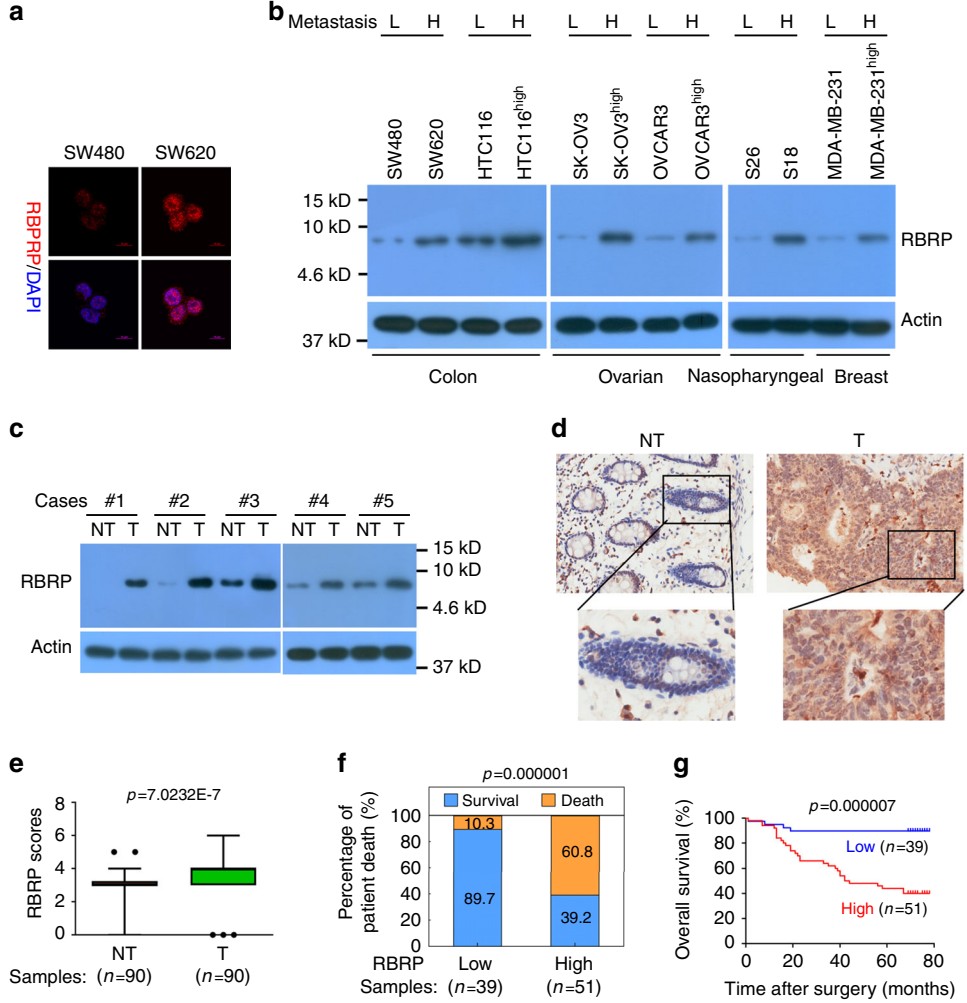

**Fig. 2 The *LINC00266-1*-encoded RBRP peptide is endogenously and naturally produced, and its upregulation is associated with a poor prognosis of patients with CRC. a** The RBRP peptide was immunostained with anti-RBRP antibodies in SW480 and SW620 cells. Scale bar: 10 μm. **b** The RBRP peptide levels were determined in the indicated cancer cell lines with different metastatic abilities (L: low; H: high). **c** Levels of the RBRP peptide were analyzed in five pairs of matched fresh primary CRC tissues (T) and their corresponding adjacent nontumoral colorectal tissues (NT). **d** Representative images of IHC staining for RBRP in CRC tissues (T) and their corresponding adjacent NT. **e** Differences in the RBRP scores between CRC tissues (T) and the corresponding adjacent NT are presented as a box plot with box & whiskers: 5–95 percentile (n = 90 tissue samples). Mann–Whitney U-test. **f** Relations between the RBRP levels and the percentage of patient death were analyzed in CRC samples (n = 39 CRC tissue samples with low RBRP and 51 samples with high RBRP). Pearson's $\chi^2$-test. **g** A Kaplan–Meier analysis of the survival of patients with CRC according to RBRP scores. Log-rank test. RBRP scores of 0–3 were low, whereas scores of 4–7 were high. Source data are provided as a Source Data file.

(Fig. 2e). Taken together, the levels of the *LINC00266-1* lncRNA and RBRP peptide were markedly increased in highly metastatic cancer cell sublines and primary cancer tissues compared with the levels in their corresponding parental cell lines and nontumoral tissues, respectively, suggesting that the lncRNA *LINC00266-1* and its encoded peptide RBRP play important roles in tumorigenesis.

**RBRP peptide[high] serves as an independent prognostic factor for patients with CRC.** The correlations between the RBRP levels and the clinicopathological features of patients with cancer were investigated in 90 pairs of matched CRC and corresponding nontumor colorectal tissue samples. Increased RBRP levels positively correlated with advanced clinical stages of CRC (p = 0.027; Supplementary Table 1). Kaplan–Meier survival analyses revealed that patients with CRC who presented with higher RBRP levels had an increased risk of cancer-related death than patients who presented with lower RBRP levels (Fig. 2f, g; p = 0.000007, log-rank test). The mean overall survival of patients with CRC

and RBRP[high] was 45.3 months, whereas the value for patients with CRC and RBRP[low] was 67.1 months. Furthermore, a multivariate Cox regression analysis indicated that RBRP[high] was an independent prognostic factor for poor survival in patients with CRC (hazard ratio = 8.26, 95% confidence interval = 2.84–24.02, p = 0.000; Supplementary Table 2). Collectively, our results reveal a significant correlation between a high RBRP level and a poor prognosis for patients with CRC, and the RBRP peptide serves as an independent prognostic factor for patients with CRC.

**Knockdown of *LINC00266-1* inhibits tumorigenesis**. We silenced *LINC00266-1* expression using two anti-*LINC00266-1* siRNAs in HCT-116 and SW620 cells to investigate the functions of *LINC00266-1* in CRC (Supplementary Fig. 2c–e). Knockdown of *LINC00266-1* inhibited HCT-116 and SW620 CRC cell proliferation, colony formation, migration, and invasion (Supplementary Fig. 5a–c). Thus, the lncRNA *LINC00266-1* is a driver of tumorigenesis.

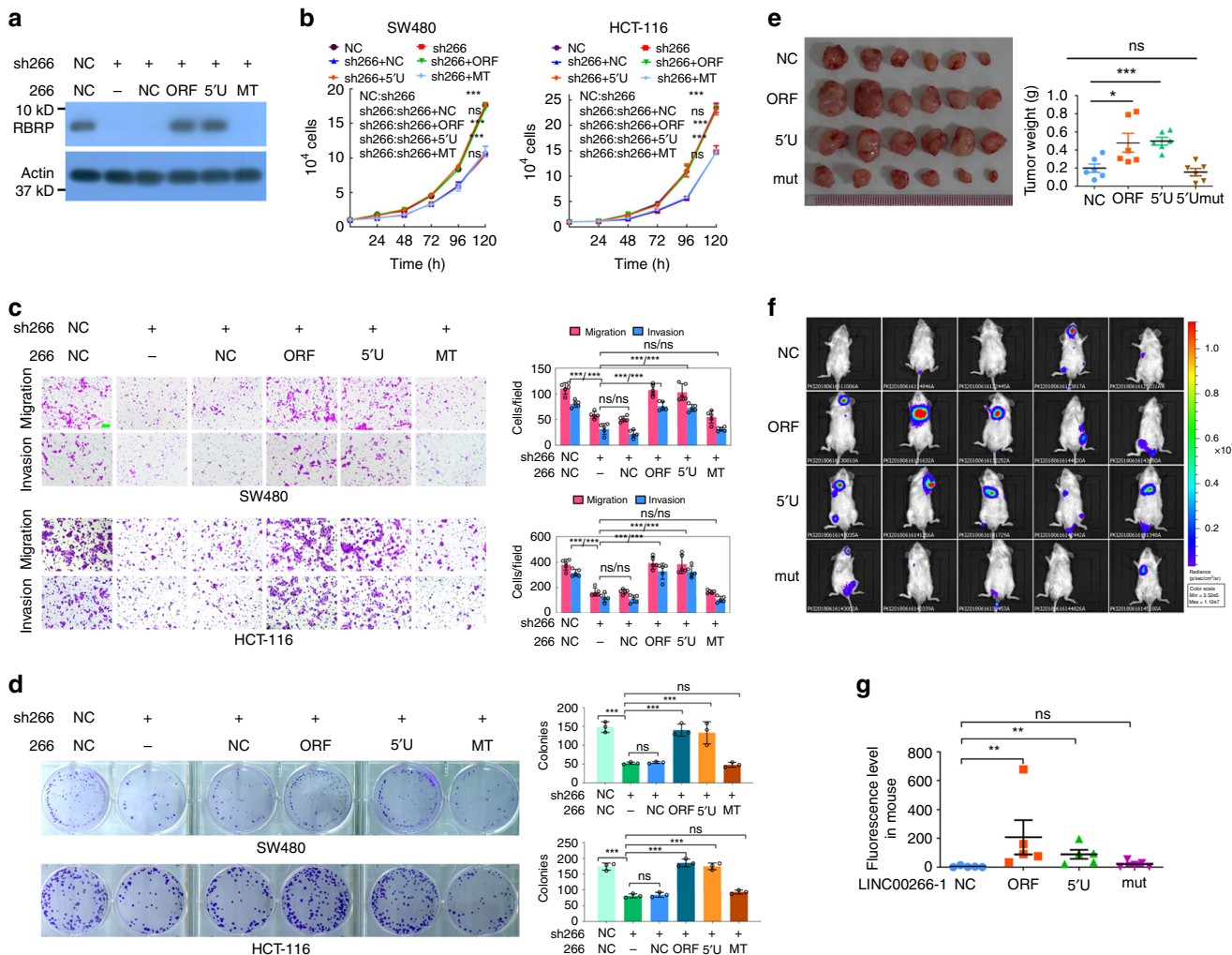

**Fig. 3 The RBRP peptide, but not the lncRNA *LINC00266-1* itself, promotes tumorigenesis and metastasis in vitro and in vivo. a–d** The indicated *LINC00266-1* (266) ORF-Flag (ORF), 5′UTR-ORF-Flag (5′U), and 5′UTR-ORFmut-Flag (MT or mut) constructs, which are resistant to anti-LINC00266-1 shRNA, were transfected into SW480 and HCT-116 cells with stable knockdown of *LINC00266-1* expression by an anti-*LINC00266-1* shRNA lentivirus (sh266); the RBRP level (**a**), cell growth (**b**) ($n = 3$ independent experiments), colony formation (**c**) ($n = 3$ independent experiments), and migration and invasion (**d**) ($n = 5$ independent experiments) were determined. Scale bar: 50 μm. **e** The in vivo tumorigenesis of the indicated cell lines stably expressing the indicated *LINC00266-1* constructs was examined. Images and weights of the xenograft tumors are provided in the left and right panels, respectively ($n = 6$ mice per group). **f** NOD-SCID mice were injected with Luc-labeled HCT-116 cells ($2 \times 10^6$ cells/mouse) stably expressing the indicated *LINC00266-1* constructs via the tail vein; luciferase activities were visualized at 9 weeks posttransplantation ($n = 5$ mice per group). **g** The fluorescence levels of mice in **f** were analyzed. Two-tailed unpaired Student's *t*-test unless specifically stated, two-way ANOVA in **b**. The data are represented as the means ± SD. *$p < 0.05$, **$p < 0.01$, or ***$p < 0.001$, ns indicates no significance. Source data are provided as a Source Data file.

**The *LINC00266-1*-encoded RBRP peptide, but not the lncRNA *LINC00266-1* itself, promotes tumorigenesis.** We used the lentivirus expressing the *LINC00266-1* short hairpin RNA (shRNA), which targets the 3′-UTR of the *LINC00266-1* ORF, to stably knock down *LINC00266-1* expression. Two CRC cell lines, HCT-116 and SW480, were constructed with the stable knockdown of *LINC00266-1*. In these cells, we restored the expression of ORF, 5′U, or MT, which confer resistance to shRNAs against *LINC00266-1* (Fig. 3a). Knockdown of *LINC00266-1* resulted in significant decreases in cell growth, colony formation, migration, and invasion, whereas these alterations were restored to the control level after the addition of ORF expression, but not *LINC00266-1* MT expression (Fig. 3b–d), indicating that the RBRP peptide, but not the lncRNA *LINC00266-1* itself, promotes tumorigenesis.

In addition, we transfected the *LINC00266-1* ORF (ORF), 5′UTR-ORF (5′U), and 5′UTR-ORFmut (mut) Flag fusion

constructs into two CRC cell lines to analyze the effects of the *LINC00266-1*-encoded RBRP peptide and the lncRNA *LINC00266-1* itself on cancer progression. The LINC00266-1 ORF and 5′UTR-ORF constructs, both of which produce the RBRP peptide, promoted CRC cell proliferation, colony formation, migration, and invasion (Supplementary Fig. 5d–f). However, the *LINC00266-1* 5′UTR-ORFmut constructs, which do not encode the RBRP peptide and function as lncRNAs, did not alter cancer cell proliferation, colony formation, migration, or invasion.

The in vivo growth of xenografts derived from cancer cells stably expressing ORF and 5′U was significantly increased, whereas ectopic stable expression of *LINC00266-1* MT constructs did not significantly alter xenograft tumor growth (Fig. 3e). In addition, larger metastatic nodules were developed in mice after tail vein injections with luciferase (Luc)-tagged cancer cells stably expressing ORF or 5′U than those in mice injected with the

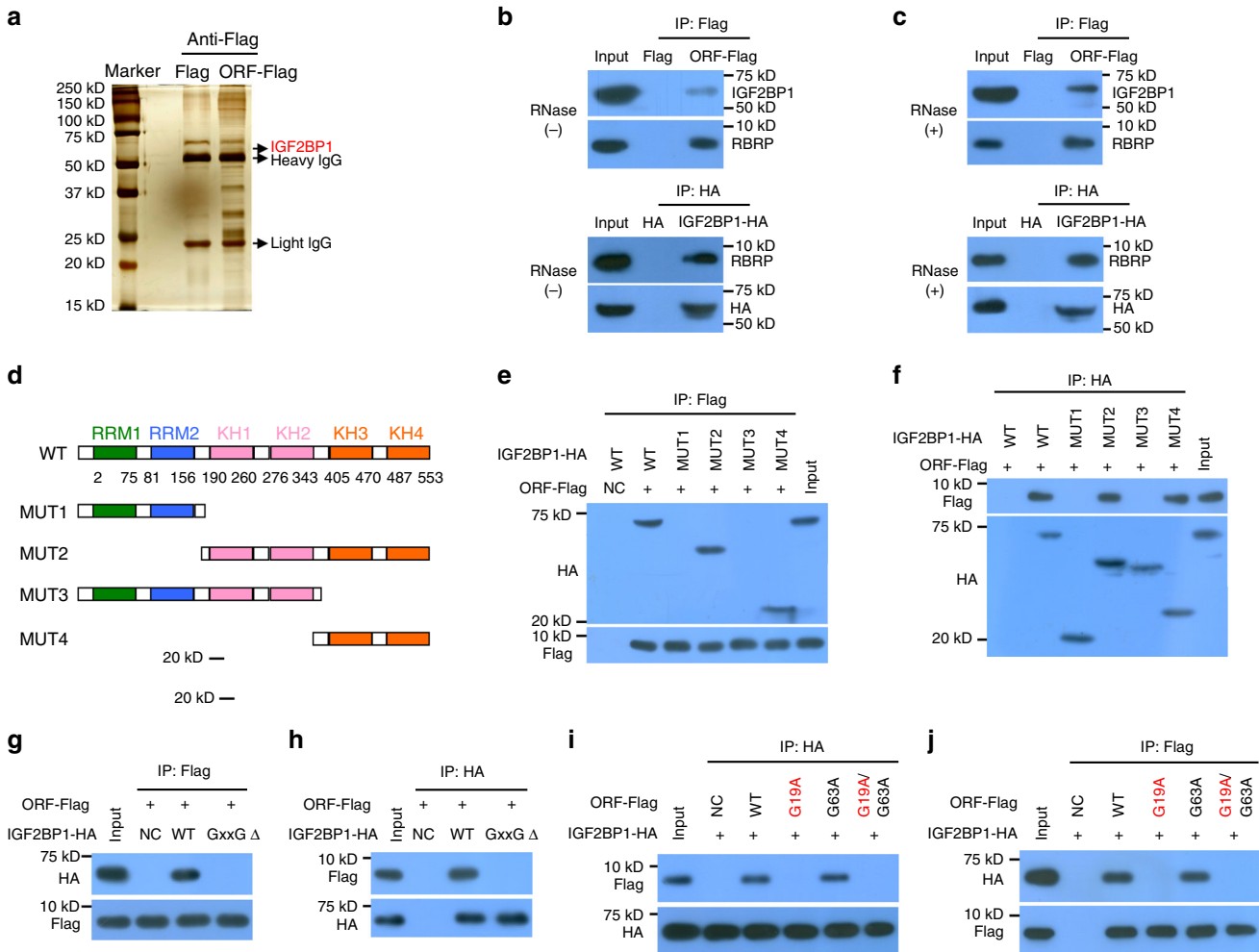

**Fig. 4 RBRP interacts with the m⁶A reader IGF2BP1. a** Proteins that interacted with RBRP were identified by co-IP together with mass spectrometry. **b**, **c** The *LINC00266-1* ORF-Flag (upper panel) and *IGF2BP1-HA* (lower panel) vectors were transfected into HEK293T cells, RBRP-Flag and IGF2BP1-HA complexes were co-IPed with anti-Flag and anti-HA antibodies in the absence (**b**) or presence (**c**) of RNase A treatment, and IGF2BP1 and RBRP/HA were detected, respectively. **d** Diagram of the different domains of the WT and mutated-*IGF2BP1* constructs. **e**, **f** The indicated *IGF2BP1-HA* mutants were cotransfected with the *LINC00266-1* ORF-Flag vector into HEK293T cells; RBRP-Flag (**e**) and IGF2BP1-HA (**f**) complexes were co-IPed with anti-Flag and anti-HA antibodies, respectively. The IGF2BP1-HA mutant and RBRP-Flag complexes were detected using anti-HA and anti-Flag antibodies, respectively. **g**, **h** The WT and GxxGΔ-mutated *IGF2BP1-HA* vectors were transfected into HEK293T cells and the interactions of RBRP with the IGF2BP1 GxxGΔ mutant were determined. **i**, **j** The WT or mutated *RBRP-Flag* constructs were cotransfected with the *IGF2BP1-HA* plasmid into HEK293T cells, and the interactions of the RBRP mutants with IGF2BP1 were determined. Source data are provided as a Source Data file.

negative control (NC) cells; however, the injection with cells stably expressing *LINC00266-1* MT did not significantly alter the metastatic nodules developed in mice (Fig. 3f, g). Collectively, the RBRP peptide produced by the lncRNA *LINC00266-1*, rather than the *LINC00266-1* lncRNA itself, promotes tumorigenesis and metastasis in vitro and in vivo, and the RBRP peptide plays oncogenic roles.

**RBRP interacts with the RNA m⁶A reader IGF2BP1**. RBRP is an uncharacterized peptide with a lack of species conservation and homology to other proteins and domains/motifs. We identified the proteins that interact with RBRP using MS to elucidate the mechanism by which RBRP promotes tumorigenesis. One hundred and seven potential interactors with RBRP were here identified by interactomics assay (Fig. 4a and Supplementary Data 2). A gene annotation assay showed that 76 of these 107 proteins belong to the RNA-binding proteins. Thus, we termed the *LINC00266-1*-encoded peptide RBRP. The RNA-binding

protein, RNA m⁶A reader IGF2BP1, was particularly interesting, because the protein score of IGF2BP1 was TOP5 in the identified RBRP interactors, RNA m⁶A modification play an important role in the hallmark of cancer, and IGF2BP1 regulates the level of an important oncogene c-Myc and tumorigenesis[6,19,20], consistent with the oncogenic roles of RBRP in tumorigenesis. Therefore, we here select IGF2BP1 for further investigation. We further confirmed that IGF2BP1 was in RBRP coimmunoprecipitated (co-IPed) complexes, and that RBRP was in IGF2BP1 co-IPed complexes, indicating that RBRP bound to IGF2BP1 (Fig. 4b). Given that IGF2BP1 binds to m⁶A-modified RNA, we further investigated whether the interaction of RBRP with IGF2BP1 depends on RNA. We found that RBRP still interacted with IGF2BP1 in the presence of RNase A (Fig. 4c), indicating that the binding of RBRP to IGF2BP1 was RNA independent.

The IGF2BP1 protein consists of two N-terminal RNA recognition domains (RRM1 and RRM2) and four C-terminal heterogeneous ribonucleoprotein (hnRNP) K homology (KH)

domains (KH1–4)[21]. We generated *IGF2BP1-HA* truncated fusion constructs (Fig. 4d) and coexpressed these constructs in HEK293T cells with ORF to investigate which domains bind to RBRP. We found that RBRP bound to the KH3–4 di-domain of IGF2BP1 (Fig. 4e, f). A previous study reported that the GxxG motif in the KH3–4 di-domain is indispensable for m6A recognition and binding[6]. We found that mutating GxxG to GEEG in the KH3–4 di-domain completely abolished the interaction of RBRP with IGF2BP1 (Fig. 4g, h).

The domain or residue of RBRP binding to IGF2BP1 was further investigated. The mutation of G19, but not G63, to A destroyed the binding of RBRP to IGF2BP1 (Fig. 4i, j), suggesting that the G19 residue in RBRP is essential for IGF2BP1 binding. Taken together, RBRP binds to the RNA m6A reader IGF2BP1.

**RBRP promotes tumorigenesis through IGF2BP1.** As expected, silencing of *IGF2BP1* suppressed CRC cell proliferation, colony formation, migration, and invasion in HCT-116 and SW480 CRC cells (Supplementary Fig. 6), similar to the cancer phenotypes induced by RBRP knockdown. To investigate whether RBRP promotes tumorigenesis through IGF2BP1, we cotransfected cells with ORF (encoding RBRP) or ORFmut-Flag (MUT) (as *LNC00266-1* lncRNA) with anti-*IGF2BP1* siRNA (Supplementary Fig. 7a). The increases in cell proliferation, colony formation, and migration and invasion induced by RBRP overexpression were completely blocked by *IGF2BP1* knockdown (Supplementary Fig. 7b–d), indicating that RBRP promotes tumorigenesis through IGF2BP1.

**RBRP strengthens the m6A recognition by the m6A reader IGF2BP1 on RNAs.** A previous study showed that IGF2BP1 is an RNA m6A reader[6]. We used m6A-methylated single-stranded RNA oligos with the consensus sequence GG(m6A)CU (ss-m6A) to investigate the influence of RBRP on m6A recognition by IGF2BP1 on RNAs. RBRP, but not the LINC00266-1 lncRNA itself, enhanced the binding of IGF2BP1 to m6A-RNA oligos, whereas RBRP did not change the unbinding of IGF2BP1 to the m6A-unmethylated control RNA (ss-A) (Supplementary Fig. 8a). The GxxG motifs in the IGF2BP1 KH3–4 di-domain are essential for m6A recognition and binding. RBRP did not increase the binding of IGF2BP1 to m6A-RNA when GxxG was mutated to GEEG (GxxGΔ) (Supplementary Fig. 8b). Moreover, the mutation of G19 to A19 in RBRP, which did not bind to the m6A reader IGF2BP1, completely destroyed the effects of RBRP on increasing the binding of endogenous IGF2BP1 to m6A-methylated RNA oligos (Supplementary Fig. 8c). Similar to endogenous RBRP peptides, a recombinant RBRP oncopeptide increased the binding of the recombinant IGF2BP1 protein to m6A-methylated RNA oligos, whereas a recombinant G19A-mutated RBRP oncopeptide did not (Supplementary Fig. 8d). Collectively, RBRP enhances the m6A recognition by IGF2BP1 on RNAs.

Previous studies have shown that IGF2BP1 stabilizes the *c-Myc* mRNA by associating with the coding region instability determinant (CRD)[20] and the m6A-seq assay showed that the CRD was marked with m6A peaks[6]. *C-Myc* is one of the most important oncogenes, consistent with the oncogenic roles of RBRP in tumorigenesis. Thus, *c-Myc* is particularly interesting and we select *c-Myc* for a systematic investigation. An ~250 nt CRD in the 3′-terminus of the *c-Myc* mRNA coding region has been reported to be critical for IGF2BP1 binding and m6A-seq showed that the CRD region has a high abundance of m6A modifications[6]. Two CRD RNA oligos containing the consensus sequence GG(m6A)CU (ss-m6A) (CRD1 and CRD2) were synthesized according to the previous study[6]. RBRP, but not

the *LINC00266-1* lncRNA itself, increased the binding of endogenous IGF2BP1 to m6A-methylated *c-Myc* CRD RNA oligos, but did not change the unbinding of IGF2BP1 to m6A-unmethylated c-Myc CRD RNA oligos (Fig. 5a). The RNA immunoprecipitation (RIP)-quantitative PCR (qPCR) assay further confirmed that RBRP, but not the *LINC00266-1* lncRNA, increased the binding of endogenous IGF2BP1 to *c-Myc* CRD RNA in cells (Fig. 5b). Similar to endogenous RBRP peptides, a recombinant RBRP oncopeptide increased the binding of the recombinant IGF2BP1 protein to the m6A-methylated *c-Myc* mRNA CDR oligos (Fig. 5f). Moreover, IGF2BP1-bound RNAs had more RNA m6A modification abundance when RBRP, but not the *LINC00266-1* lncRNA itself, was overexpressed in cells (Fig. 5c). RBRP did not increase the recognition by endogenous IGF2BP1 for the m6A-modified mRNA CRD of *c-Myc* when the GxxG motifs in the IGF2BP1 KH3–4 di-domain were mutated to GEEG (GxxGΔ) (Supplementary Fig. 9a, b). These data indicate that RBRP increases the binding of IGF2BP1 to the m6A-modified mRNA CRD of *c-Myc*.

Furthermore, we found that the RBRP G19A mutant, which does not bind to IGF2BP1, did not increase the binding of endogenous IGF2BP1 to m6A-methylated *c-Myc* CRD RNA oligos (Fig. 5d). An RIP-qPCR assay also demonstrated that the RBRP G19A mutant did not enhance the binding of endogenous IGF2BP1 to *c-Myc* CRD RNA in cells (Fig. 5e). In addition, a recombinant G19A-mutated RBRP oncopeptide did not increase the binding of the recombinant IGF2BP1 protein to the m6A-methylated *c-Myc* mRNA CDR oligos (Fig. 5f). IGF2BP1-bound RNAs in wild-type (WT) RBRP-expressed cells had more RNA m6A modification abundance than those in RBRP G19A mutant-expressed cells (Fig. 5g). These results indicate that RBRP increases the binding of IGF2BP1 to m6A-modified *c-Myc* CRD mRNA by its G19 residue.

As shown above, RBRP did not increase the in vitro binding of IGF2BP1 to m6A-unmethylated RNA oligos. An RIP-qPCR assay further demonstrated that the enhancement of the binding of endogenous IGF2BP1 to the *c-Myc* CRD RNA induced by RBRP overexpression was significantly impaired when the cellular RNA m6A level was decreased by silencing the expression of the m6A writer METTL14 (Fig. 5h and Supplementary Fig. 9c). Moreover, we generated a *c-Myc* CRD mutant (CRDmut) in which A was mutated to U within six m6A consensus sequences in *c-Myc* CRD. RIP-qPCR and m6A-PCR assays demonstrated that RBRP did not increase the recognition of IGF2BP1 to m6A-modified *c-Myc* CRD mRNA when A was mutated to U within the six m6A consensus sequences in *c-Myc* CRD in cells (Fig. 5i and Supplementary Fig. 9d), indicating that RBRP increases the recognition and binding of IGF2BP1 to *c-Myc* CRD mRNA in an m6A-dependent manner. Taken together, RBRP, but not the lncRNA *LINC00266-1* itself, strengthened the m6A recognition and binding of the m6A reader IGF2BP1 on RNAs, such as *c-Myc* mRNA.

**RBRP increases *c-Myc* mRNA stability by strengthening the m6A recognition by IGF2BP1 on *c-Myc* mRNA.** A previous study showed that IGF2BP proteins enhance the stability of target mRNAs in an m6A-dependent manner[6]. Thus, we further investigated the influences of RBRP on *c-Myc* mRNA stability. RBRP, not the lncRNA *LINC00266-1* itself, increases the stability of *c-Myc* mRNA, as well as the *c-Myc* mRNA and protein levels (Fig. 6a–c). The silencing of IGF2BP1 completely blocked the enhancement of *c-Myc* mRNA stability induced by RBRP overexpression (Fig. 6d), indicating that RBRP increased *c-Myc* mRNA stability mainly through the m6A reader IGF2BP1.

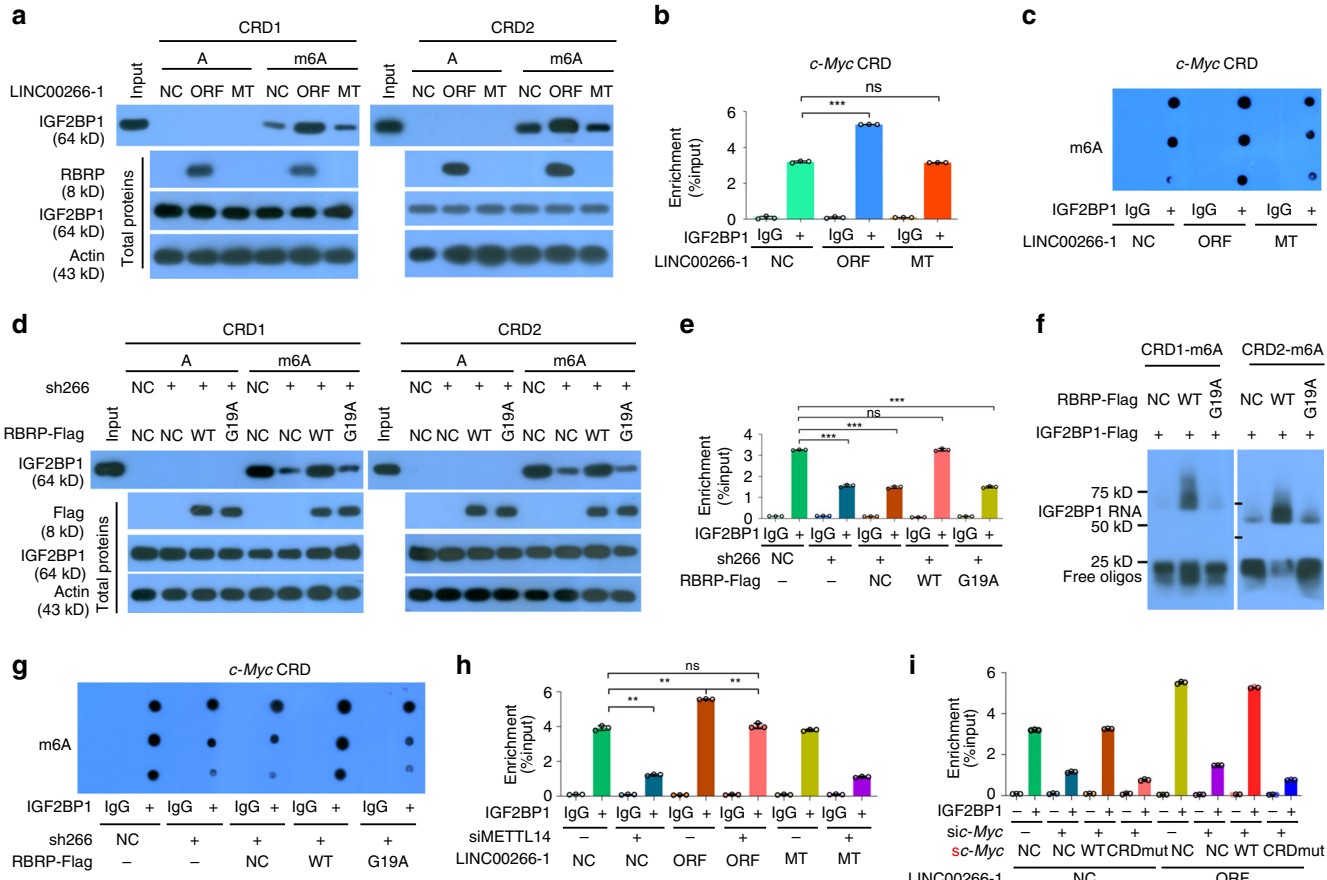

**Fig. 5 The RBRP oncopeptide, not the lncRNA *LINC00266-1* itself, strengthens the recognition and binding of the m⁶A reader IGF2BP1 to m⁶A-modified c-Myc CRD mRNA. a** The in vitro binding of IGF2BP1 to m⁶A-unmethylated or -methylated *c-Myc* CRD mRNA oligos was investigated in HCT-116 cells stably expressing the *LINC00266-1* ORF or 5′UTR-ORFmut (MT) by RNA pull-down assays. **b** The in vivo binding of IGF2BP1 on *c-Myc* CRD mRNA was determined in cells as in **a** by an RIP-qPCR assay (*n* = 3 independent experiments). **c** The m⁶A levels in IGF2BP1-bound RNAs were detected in cells as in **a** by dot blotting using an anti-m⁶A antibody. **d** WT or G19A-mutated *RBRP-Flag* vectors were transfected into HCT-116 cells with stable knockdown of *LINC00266-1* expression (sh266); the in vitro binding of IGF2BP1 to m⁶A-unmethylated or -methylated *c-Myc* CRD mRNA oligos was analyzed by RNA pull-down assays. **e** The in vivo binding of IGF2BP1 on *c-Myc* CRD mRNA was determined in cells treated as in **d** by an RIP-qPCR assay (*n* = 3 independent experiments). **f** Recombinant IGF2BP1 protein, WT, or G19A-mutated RBRP peptide and m⁶A-methylated *c-Myc* CRD mRNA oligos were incubated, and the binding capability of IGF2BP1 on m⁶A-methylated *c-Myc* CRD mRNA was analyzed by an RNA EMSA assay. **g** The m⁶A levels in IGF2BP1-bound RNAs were detected in cells treated as in **d** by dot blotting. **h** The m⁶A writer *METTL14* expression was silenced in HCT-116 cells stably expressing the *LINC00266-1* ORF or 5′UTR-ORFmut (MT); the in vivo binding of IGF2BP1 on *c-Myc* CRD mRNA was determined by an RIP-qPCR assay (*n* = 3 independent experiments). **i** The WT or CRD-mutated *c-Myc* plasmids (sc-Myc), which were also synonymously mutated and resistant to anti-c-Myc siRNA, were cotransfected together with anti-c-Myc siRNAs into HCT-116 cells stably expressing the *LINC00266-1* ORF; the in vivo binding of IGF2BP1 to *c-Myc* CRD mRNA was determined by an RIP-qPCR assay (*n* = 3 independent experiments). Two-tailed unpaired Student's *t*-test. The data are represented as the means ± SD. \**p* < 0.05, \*\**p* < 0.01, or \*\*\**p* < 0.001, ns indicates no significance. Source data are provided as a Source Data file.

Furthermore, the enhancement of the stability of *c-Myc* mRNA induced by RBRP overexpression was abolished when the m⁶A writer METTL14 was knocked down (Fig. 6e), suggesting that RBRP-mediated *c-Myc* mRNA stability is dependent on mRNA m⁶A modification. The A to U mutations within the six m⁶A consensus sites in *c-Myc* CRD mRNA significantly abrogated the enhanced stability of *c-Myc* mRNA and the increased *c-Myc* mRNA and protein levels induced by RBRP overexpression in cells (Fig. 6f–h). These results show that RBRP increases *c-Myc* mRNA stability in a *c-Myc* CRD mRNA m⁶A-dependent manner.

We next sought to determine whether the RBRP oncopeptide level and *c-Myc* level are correlated in clinical samples. The *c-Myc* mRNA and protein levels in CRC tissues were higher than those in the corresponding nontumoral colorectal tissues (Fig. 6i, j). The *c-Myc* levels were positively correlated with RBRP oncopeptide levels in clinical sample tissues ($R^2 = 0.5921$, *p* = 0.0093) (Fig. 6k).

The RNA stabilizers HuR, MATR3, and PABPC1 have been identified to regulate the stability of target mRNAs[22–24]. We demonstrated that RBRP increased the interaction of IGF2BP1 with the RNA stabilizers HuR, MATR3, and PABPC1 in an RNA-independent manner, whereas the mutation of G19A in RBRP, which did not bind to IGF2BP1, did not increase the binding of IGF2BP1 to these RNA stabilizers (Supplementary Fig. 10a, b). Moreover, RBRP enhanced the binding of these RNA stabilizers to m⁶A-RNA probes and m⁶A-methylated *c-Myc* CRD RNAs but did not promote the binding of these RNA stabilizers to m⁶A-unmethylated RNA probes and m⁶A-unmethylated *c-Myc* CRD RNAs (Supplementary Fig. 10c, d). However, the RBRP G19A mutant, which does not bind to IGF2BP1, did not increase the binding of these RNA stabilizers to m⁶A-methylated *c-Myc* CRD RNAs (Supplementary Fig. 10e). Thus, RBRP strengthens the recruitment of the RNA stabilizers HuR, MATR3, and PABPC1 to m⁶A-*c-Myc* CRD RNAs, to promote the stability of *c-Myc*

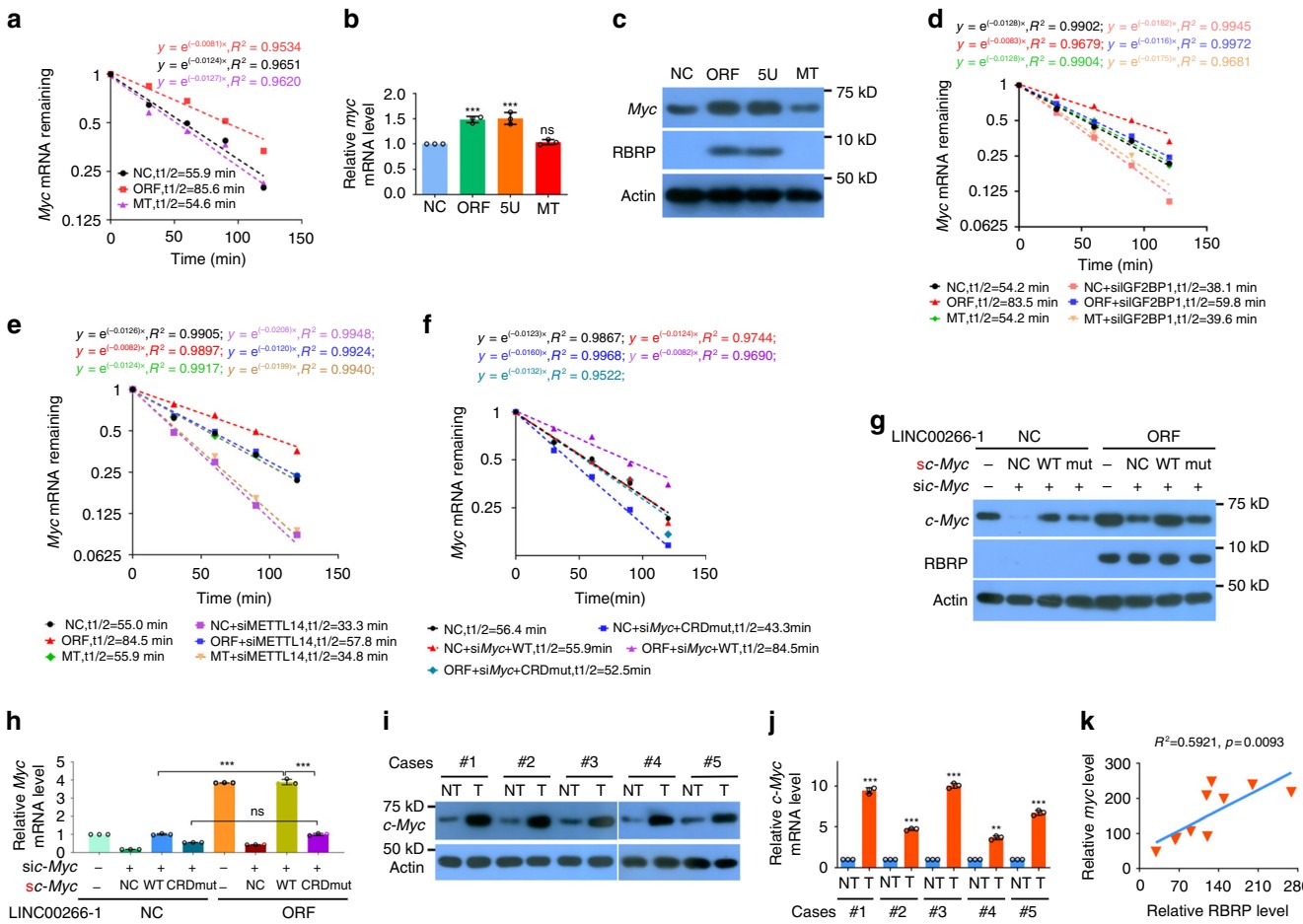

**Fig. 6 RBRP increases the stability and expression of *c-Myc* mRNA by regulating the m⁶A recognition by IGF2BP1 on *c-Myc* CRD mRNA. a–c** RBRP overexpression, but not the lncRNA *LINC00266-1* itself, increases the half-life (**a**) (*n* = an experiment) and level (**b**) (*n* = 3 independent experiments) of *c-Myc* mRNA and the c-Myc protein level (**c**) in the cells stated in Fig. 5a. **d, e** The enhancement of the *c-Myc* mRNA half-life induced by RBRP (ORF) was blocked by silencing either the m⁶A reader IGF2BP1 (**d**) or the m⁶A writer METTL14 (**e**) in the cells stated in Fig. 5a (*n* = an experiment). **f–h** Cells were treated as in Fig. 5h. RBRP overexpression did not increase the *c-Myc* mRNA half-life (**f**) (*n* = an experiment) or level (**h**) (*n* = 3 independent experiments), or the c-Myc protein level (**g**) when A was mutated to U within six m⁶A consensus sites in *c-Myc* mRNA CRD. **i, j** The *c-Myc* protein (**i**) and mRNA (**j**) (*n* = 3 independent experiments) levels were determined in the CRC tissue samples shown in Fig. 2c. The *c-Myc* protein and mRNA levels were increased in CRC tissues (T) compared with those in the corresponding NT. **k** The *c-Myc* protein levels were positively correlated with the RBRP oncopeptide levels in the clinical tissue samples (*n* = 10 samples) using linear regression analysis. Two-tailed unpaired Student's *t*-test unless specifically stated. The data are represented as the means ± SD. *$p < 0.05$, **$p < 0.01$, or ***$p < 0.001$, ns indicates no significance. Source data are provided as a Source Data file.

mRNA and upregulate the *c-Myc* levels. Collectively, our results indicated that RBRP increases *c-Myc* mRNA stability by strengthening the m⁶A recognition by IGF2BP1 on *c-Myc* CRD mRNA and the recruitment of RNA stabilizers to m⁶A-methylated-*c-Myc* CRD mRNA.

**RBRP promotes tumorigenesis through c-Myc.** *C-Myc* is a key, well-known oncogene. We next investigated whether RBRP promoted tumorigenesis through c-Myc, because RBRP drives m⁶A recognition-dependent *c-Myc* mRNA stability. Either the ORF (encoding the RBRP oncopeptide) or MT (MUT) (*LNC00266-1* lncRNA) construct was cotransfected together with anti-*c-Myc* siRNAs into cells (Supplementary Fig. 11a). The increases in cell proliferation, colony formation, and migration and invasion induced by RBRP overexpression were completely blocked by *c-Myc* knockdown (Supplementary Fig. 11b–d). Thus, RBRP promotes tumorigenesis mainly through c-Myc.

**RBRP exerts its oncogenic roles by binding to IGF2BP1.** As shown above, RBRP increases the m⁶A recognition by IGF2BP1

on RNAs and promotes *c-Myc* mRNA stability and tumorigenesis through IGF2BP1. Therefore, we further examined whether RBRP promotes tumorigenesis and *c-Myc* mRNA stability in an IGF2BP1-binding-dependent manner. The expression of the WT RBRP or the IGF2BP1-binding-defective mutant RBRP G19A, which is resistant to the *LINC00266-1* shRNA that targeted the 3′-UTR of the RBRP ORF, was restored in CRC cells with stable knockdown of *LINC00266-1*. Knockdown of *LINC00266-1* significantly decreased the *c-Myc* mRNA stability and level, cell growth, colony formation, migration, and invasion, whereas these alterations were restored to the control levels in cells cotransfected with WT RBRP but not with the IGF2BP1-binding-defective mutant RBRP G19A (Fig. 7), indicating that RBRP promotes tumorigenesis and *c-Myc* mRNA stability by binding to IGF2BP1.

## Discussion

m⁶A RNA modifications play an important role in post-transcriptional gene regulation, development, and diseases, such as cancers[1,2]. m⁶A RNA modifications exert their biological

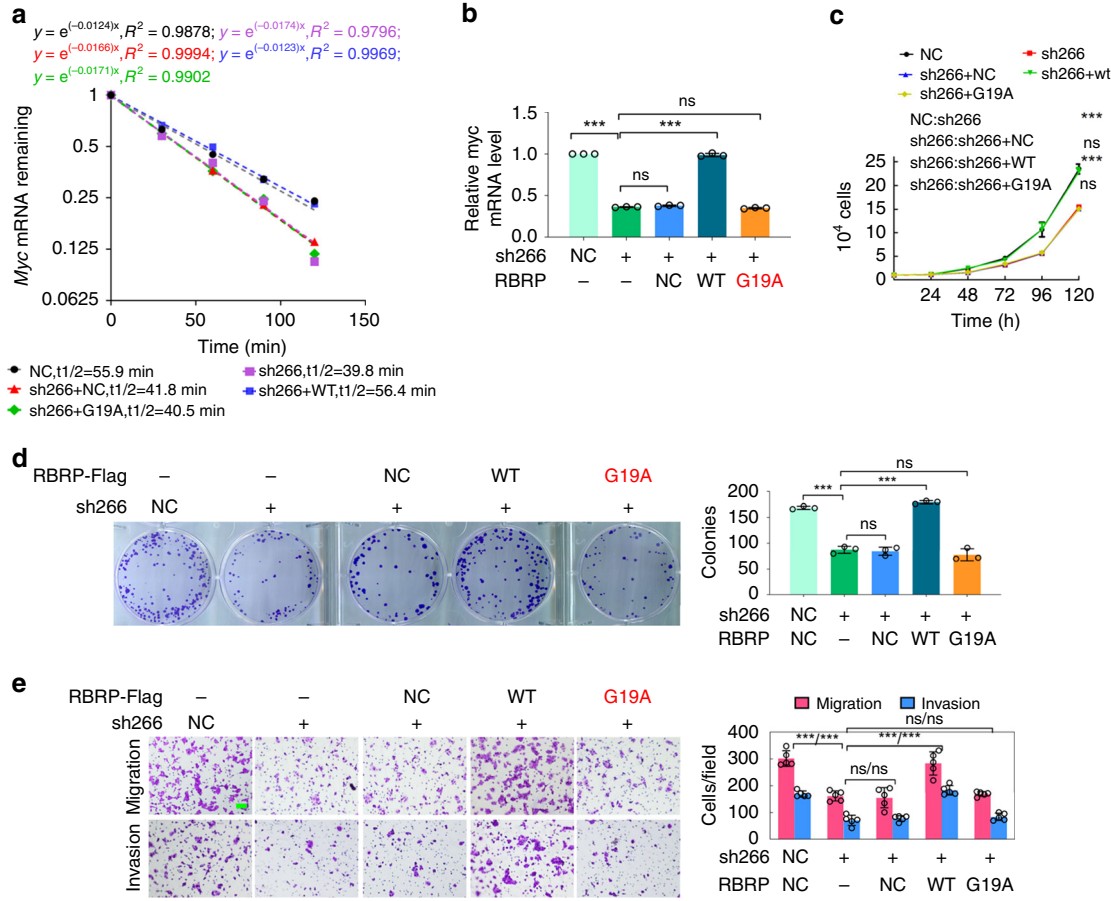

**Fig. 7 RBRP regulates *c-Myc* mRNA stability and tumorigenesis by binding to IGF2BP1. a–e** The WT or G19A-mutated *RBRP* constructs, which were resistant to anti-*LINC00266-1* shRNA, were transfected into HCT-116 cells with stable knockdown of *LINC00266-1* expression by an anti-*LINC00266-1* shRNA lentivirus targeting the 3′-UTR of the *LINC00266-1* ORF (sh266); the half-life (**a**) (*n* = an experiment) and level (**b**) (*n* = 3 independent experiments) of *c-Myc* mRNA were detected, along with cell proliferation (**c**) (*n* = 3 independent experiments), colony formation (**d**) (*n* = 3 independent experiments), and migration and invasion (**e**) (*n* = 5 independent experiments). Scale bar: 50 μm. The G19A mutation of RBRP, which did not bind to the m[6]A reader IGF2BP1, abolished the stimulatory effects of RBRP on *c-Myc* stability and expression and cancer cell proliferation, colony formation, migration, and invasion. Two-tailed unpaired Student's *t*-test unless specifically stated, two-way ANOVA in **c**. The data are represented as the means ± SD. *$p < 0.05$, **$p < 0.01$, or ***$p < 0.001$, ns indicates no significance. Source data are provided as a Source Data file.

importance through m[6]A readers. However, how these m[6]A readers recognize their target transcripts remains largely unknown. In this study, we found that an uncharacterized oncopeptide RBRP encoded by the previously annotated, uncharacterized lncRNA *LINC00266-1* is significantly upregulated in cancers and strengthens the recognition and binding of the m[6]A reader IGF2BP1 to target mRNAs, such as *c-Myc* mRNA, to increase the stability and level of *c-Myc* mRNA, thereby promoting tumorigenesis.

The effectors in the m[6]A RNA pathway include the writers METTL3 and METTL14; the erasers FTO and ALKBH5; and the readers YTHDF1-3, YTHDC1-2, IGF2BPs, and hnRNPs[1,2,4–8]. The m[6]A RNA modification is installed mainly through METTL3 and METTL14, which are the catalytic subunit and an essential component to facilitate RNA binding, respectively[3]. In addition to METTL3 and METTL14, recent studies have further shown that a handful of additional regulatory subunits, including WTAP, VIRMA, ZC3H13, HAKAI, and RBM15/15B, also play an important role in the installation of m[6]A on RNA[9–12]. However, the additional regulatory subunits of RNA m[6]A readers and m6A demethylases are not reported. In this study, our findings indicate that m[6]A readers also contain additional regulatory subunits. Here we found that the oncopeptide RBRP encoded by the

lncRNA *LINC00266-1* is a regulatory subunit of the RNA m[6]A reader IGF2BP1 and regulates the m[6]A recognition by IGF2BP1 on target RNAs. Our findings unveil an additional regulatory layer for the m[6]A pathway and m[6]A recognition.

m[6]A RNA methylation is emerging as a pathway affecting the hallmarks in a variety of cancers[2]. m[6]A RNA writers, erasers, and readers were identified to be dysregulated in several cancers and were found to regulate cancer initiation, proliferation, stemness, metastasis, resistance to chemo-/radiotherapy, and tumorigenesis in an m[6]A-dependent manner[25–30]. Although the regulatory subunits of the m[6]A writers WTAP, VIRMA, ZC3H13, and HAKAI have been reported to be associated with cancers[31–33], whether their functional roles in cancers are dependent on m[6]A modification is not reported. In this study, we found that the regulatory subunit RBRP of the m[6]A reader promotes cancer cell proliferation, colony formation, tumorigenesis, and metastasis in an m[6]A- and m[6]A reader-dependent manner. RBRP strengthens the m[6]A recognition on target mRNAs, such as oncogene *c-Myc* mRNAs by IGF2BP1 to increase the stability and level of the *c-Myc* mRNA, thereby promoting tumorigenesis. The enhancement effects of RBRP on m[6]A recognition, mRNA stability, and tumorigenesis are impaired when the m[6]A modification is absent from cellular *c-Myc* CRD mRNA. Our results indicate that, in

addition to the aberrant m⁶A modification being closely associated with cancer, the aberrant m⁶A recognition mediated by the regulatory subunit of the m⁶A reader is also closely associated with cancer.

The RBRP oncopeptide is an uncharacterized peptide that is encoded by the previously annotated lncRNA *LINC00266-1* with unknown functions. The interactomics and MS analyses identified that a total of 107 proteins interacted with the *LINC00266-1*-encoded peptide. A gene annotation assay showed that 76 of these 107 proteins belong to the RNA-binding proteins. Thus, we termed the *LINC00266-1*-encoded peptide RBRP. The RBRP peptide, not the lncRNA *LINC00266-1* itself, exerts its oncogenic functions to promote cancer cell proliferation, colony formation, migration, and invasion in vitro and to promote tumorigenesis and metastasis in vivo. Therefore, RBRP is further termed an oncopeptide. The inhibition of RBRP can significantly suppress cancer cell proliferation, colony formation, migration, and invasion. m⁶A RNA writers, erasers, and readers have been suggested to be anticancer therapeutic targets[1,34]. Our results showed that the regulatory subunit RBRP oncopeptide of the m⁶A reader may be a potential cancer prognosis biomarker and therapeutic target.

In addition to the role of RBRP in the m⁶A recognition by interacting with IGF2BP1, the gene annotation assay demonstrated that RBRP interactors mainly involve in the regulation of RNA splicing and many splicing regulators interact with RBPR including ten hnRNP family proteins, seven splicing factors, and two small nuclear ribonucleoproteins, most of which have been reported to be involved in the regulation of the hallmarks of cancer[17,35,36], suggesting that RBRP may also regulate pre-mRNA splicing to regulate tumorigenesis and cancer progression. Therefore, the interactions of RBRP with other interactors will be valuable to explore in next step.

To date, more than 28,000 lncRNA transcripts have been identified in the human genome. LncRNAs are emerging as important regulatory molecules of gene expression and have been functionally implicated in the regulation of multiple cellular processes as RNA molecules. The dysregulation of lncRNAs contributes to various cancer hallmarks[16–18]. However, until now, no lncRNA has been reported to participate in the installation, removal, or recognition of m⁶A on RNAs. In our present study, our findings reveal that the lncRNA *LINC00266-1* participates in the m⁶A recognition by m⁶A readers and the lncRNA *LINC00266-1* exerts its m⁶A recognition regulation by encoding an oncopeptide instead of an RNA molecule. Our and previous studies showed that a limited number of lncRNAs and circRNAs could encode proteins/peptides to regulate tumorigenesis[37–40]. Our findings further confirmed that some peptides are hidden in some previously annotated lncRNAs. Our report enriches and broadens the breadth and diversity of noncoding RNA functions and mechanisms in the m⁶A RNA pathway and in tumorigenesis.

In summary, a previously annotated lncRNA, *LINC00266-1*, actually encodes an uncharacterized peptide, RBRP, which exhibits oncogenic roles in tumorigenesis; this finding highlights the therapeutic potential of targeting RBRP in cancers. The RBRP oncopeptide, as an additional regulatory subunit of m⁶A readers, strengthens the m⁶A recognition on RNAs, such as *c-Myc* mRNA, by the m⁶A reader IGF2BP1 to increase *c-Myc* mRNA stability and expression. Our study uncovers a connection between an lncRNA and its encoded peptide and the m⁶A RNA pathway.

## Methods

**Cell culture and tissue samples.** SW480, SW620, HCT-116, MBA-MD-231, SK-OV-3, OVAR-3, HeLa, and HEK293T cell lines were purchased from the American Type Culture Collection and cultured under standard conditions. HCT-116^high, MDA-MB-231^high, OVCAR-3^high, and SK-OV-3^high cell sublines with increased metastatic abilities were previously established by our group[17,41]. S18 and S26 cell

lines were kindly provided as described in our previous study[17]. Cells were regularly monitored for mycoplasma contamination.

Matched fresh frozen primary CRC tissues and their corresponding adjacent nontumor samples of colorectal tissues were collected from patients with CRC at Sun Yat-sen University Cancer Center. These cases were selected based on a clear pathological diagnosis, and the patients were not preoperatively treated with anticancer agents. The collection of these tissue samples was approved by the Internal Review and Ethics Boards at the Third Affiliated Hospital of Guangzhou Medicine University. Informed consent was obtained from each patient.

Tissue microarray chips containing 90 pairs of matched CRC tissue samples, their corresponding adjacent nontumor colorectal tissue samples, and the related clinicopathological data were purchased from Shanghai OUTDO Biotech Co., Ltd (Shanghai).

**Anti-RBRP antibody production.** The epitope peptide was synthesized and anti-RBRP antibodies were produced by GL Biochem (Shanghai), Ltd. Briefly, a KLH-coupled peptide, Cys-TDTKKDKHPDPY (CY-13), was synthesized and polyclonal antibodies against RBRP were obtained from inoculated rabbits. Antibodies were further purified using affinity chromatography on columns containing the corresponding epitope peptides.

**Plasmid constructs.** The *LINC00266-1* and other lncRNA ORF-GFPmut, ORF, 5′ U, and MT, IGF2BP1-HA, IGF2BP1-Flag, *c-Myc*-Flag plasmids were generated as previously described[17]. The *IGF2BP1-HA* MUT1, MUT2, MUT3, MUT4, GxxGΔ (GxxG mutates to GEEG) mutants, RBRP-Flag G19A, G63A, and G19A/G63A mutants, *c-Myc* CRDmut mutant, and synonymous mutated IGF2BP1 and *c-Myc* mutants were generated using a Mut Express II Fast Mutagenesis Kit V2 (Vazyme, China). The ORF, 5′U, and MT sequences were provided in Supplementary Table 3.

**RNA interference.** siRNAs (60 nM) against LINC00266-1, IGF2BP1, or *c-Myc* genes or NC siRNA (GenePharma) were transfected into the cells with RNAiMAX (Invitrogen) for 48 h (unless otherwise stated). The plasmids together with 50 nM siRNAs were cotransfected using Lipofectamine 2000 (Invitrogen) for 48 h (unless otherwise stated). The siRNA sequences were provided in Supplementary Table 3.

**Immunofluorescence staining.** HeLa cells were transfected with *LINC00266-1* and other lncRNA ORF-GFPmut vectors, and GFP fluorescence was directly visualized and recorded. HeLa cells were transfected with the indicated plasmids or the indicated siRNAs for 24 h, and these cells were then plated and cultured on glass cover slips for 24 h. These cells were fixed with 4% paraformaldehyde, permeabilized with 0.1% Triton X-100, and incubated with anti-RBRP or anti-Flag antibodies at 4 °C overnight. Subsequently, these cells were treated with the corresponding Alexa Fluor 488- or Cy3-conjugated secondary IgG antibodies (1:1000), and cellular nuclei were stained with DAPI. The images were obtained by laser scanning confocal microscopy.

**Western blotting.** Whole-cell lysates and tissue lysates were prepared, and the protein concentrations were determined as previously described[42]. Whole-cell lysates and tissue lysates were separated using 10–12.5% Tris-SDS-polyacrylamide gel electrophoresis (PAGE) and then electroblotted onto a polyvinylidene fluoride (PVDF) membrane. The indicated proteins were detected using anti-RBRP (our developed, 1:500), Flag (M185-3L, MBL, RRID: AB_11123930, 1:2000), HA (561, MBL, RRID: AB_591839, 1:2000), GFP (50430-2-AP, Proteintech, RRID: AB_11042881, 1:1000), IGF2BP1 (8482, CST, 1:1500), *c-Myc* (13987, CST, RRID: AB_11179079, 1:1000), m6A (ABE572, Merck Millipore, 1:2000), HuR (11910-1-AP, Proteintech, RRID: AB_11182183, 1:2000), PABPC1 (10970-1-AP, Proteintech, RRID: AB_10596918, 1:2000), MATR3 (12202-2-AP, Proteintech, RRID: AB_2281752, 1:2000), and β-actin (60008-1-Ig, Proteintech, RRID: AB_2289225, 1:4000) antibodies.

**Tricine-SDS-PAGE and low-molecular-weight peptide detection.** Whole-cell lysates and tissue lysates were prepared and the protein concentrations were determined as previously described[42]. Whole-cell lysates and tissue lysates were separated on 16% Tricine-SDS-PAGE gels as previously described[17,43] and then electroblotted onto a PVDF membrane. Western blotting was performed using anti-Flag (M185-3L, MBL, RRID: AB_11123930) or anti-RBRP (1:500) antibodies to detect RBRP oncopeptide.

**RT-PCR and qRT-PCR.** Total RNA or RIP RNA was extracted using TRIzol total RNA isolation reagent (Invitrogen). *LINC00266-1* lncRNA, IGF2BP1, METLL14, *c-Myc*, *c-Myc* CRD, and *GAPDH* mRNA levels were determined using reverse transcriptase PCR (RT-PCR) or quantitative RT-PCR (qRT-PCR). The sequences of the RT-PCR and qRT-PCR primers were provided in Supplementary Table 3.

**IHC assay.** IHC assays were performed as previously described using anti-RBRP antibodies[44]. IHC assays were performed on colon cancer tissue microarray chips

with anti-RBRP antibody (1:600). All IHC results were assessed by two independent pathologists blinded to both the sample origins and the subject outcomes. RBRP oncopeptide level scores were evaluated using a previously described semi-quantitative German scoring system with the percentage of positive cells and the staining intensity. Scores of 0–3 indicated low RBRP micropeptide levels (low) in cancer tissues, whereas scores of 4–7 indicated high RBRP micropeptide levels (high) in cancer tissues.

**Cell proliferation assay**. Cell proliferation assays were performed as previously described[17,41]. Briefly, cells were transfected with the indicated siRNAs and/or plasmids for 12 h. These cells were suspended, and $1 \times 10^4$ cells were then plated in 96-well culture plates and cultured in culture medium supplemented with 10% fetal bovine serum (FBS). The cell number was counted at 24, 48, 72, 96, and 120 h.

**Colony formation assays**. Cell proliferation assays were performed as previously described[17,41]. Briefly, cells were transfected with the indicated siRNAs and/or plasmids for 12 h. Then, 250 cells were plated in six-well culture plates and cultured in medium supplemented with 10% FBS. These cells were then fixed with methanol and stained with crystal violet solution. The colony numbers were counted under a microscope ($n = 3$).

**Migration and invasion assays**. In vitro migration and invasion assays were performed using Transwell chambers as previously described[45]. Cells were transfected with the indicated siRNAs or/and plasmids. A total of $1 \times 10^5$ cells in 100 μL cell suspensions with 0.05% FBS was added to the upper transwell chambers for the migration assay (8.0 μM pore size, BD) or the upper transwell chambers coated with Matrigel for the invasion assay, and medium with 10% FBS was added to the bottom chamber. Migrated and invasive cells were stained with 5% crystal violet. Images were obtained from each membrane, and migrated and invasive cells were counted under a microscope.

**Construction of cell lines with stable expression**. The HCT-116-Luc cell line stably expressing Luc was previously established by our group[17]. HCT-116-Luc cells were further infected with lentiviruses expressing the indicated LINC00266-1 constructs and selected using puromycin. The ectopic expression of the RBRP-Flag fusion protein was validated by western blotting. HCT-116 cell lines stably expressing LINC00266-1-ORF-Flag (HCT-116-Luc-LINC00266-1-ORF-Flag), LINC00266-1-5′U (HCT-116-Luc- LINC00266-1-5′UTR-ORF-Flag), or LINC00266-1-MT (HCT-116-Luc- LINC00266-1-MT) were established.

HCT-116 cells were infected with lentiviruses expressing LINC00266-1 shRNA targeted to the 3′-UTR region of LINC00266-1 ORF and selected using puromycin. HCT-116 cell lines stably expressing LINC00266-1 shRNA were established.

**Animal studies**. Male BALB/c nude mice (3–4 weeks old) were purchased from Charles River Laboratories in China (Beijing). An in vivo tumorigenesis assay was performed as previously described[17,41]. Briefly, $1 \times 10^6$ HCT-116-Luc-LINC00266-1-ORF-Flag, HCT-116-Luc- LINC00266-1-5′U, or HCT-116-Luc-LINC00266-1-MT cells were subcutaneously injected into the dorsal flanks of each BALB/c nude mouse ($n = 6$). After 3 weeks, the mice were killed and the tumors were dissected and weighed.

Male NOD-SCID mice (3–4 weeks old) were obtained from Charles River Laboratories in China. An in vivo metastasis assay was performed using previously described methods[17,41]. Briefly, NOD-SCID mice in each experimental group were injected with $2 \times 10^6$ HCT-116-Luc-LINC00266-1-ORF-Flag, HCT-116-Luc-LINC00266-1-5′U, or HCT-116-Luc-LINC00266-1-MT cells through the tail vein ($n = 5$). The metastatic foci were visualized 9 weeks after implantation using an IVIS 200 Imaging System (Xenogen).

The mice used in these experiments were bred and maintained under defined conditions at the Animal Experiment Center of the College of Medicine (SPF-grade facility), Jinan University, in which ambient temperature and humidity is $22 \pm 2$°C and 40–70%, respectively. The animal experiments were approved by the Laboratory Animal Ethics Committee of The Third Affiliated Hospital of Guangzhou Medicine University and Jinan University and conformed to the legal mandates and national guidelines for the care and maintenance of laboratory animals.

**Coimmunoprecipitation assay**. Cells were transfected with the indicated plasmids. Whole-cell lysates from cells were prepared as previously reported[42]. Coimmunoprecipitation assay (Co-IP) was performed using anti-Flag or anti-HA antibodies, and the immune complexes were captured on Protein A/G agarose beads (Santa Cruz). The co-IPed complexes were separated. The gels were stained with silver for MS assay or used for western blotting assay with the indicated antibodies. For the MS assay, three independent experiments were performed; the differential gel bands and their corresponding negative gel bands were excised and were in-gel-digested with trypsin.

**MS assay**. For identification of exogenous RBRP peptide, the LINC00266-1 ORF-GFPmut plasmid was transfected into HeLa cells, the RBRP-GFP fusion peptide

was immunoprecipitated using anti-GFP antibody, the immunoprecipitated RBRP-GFP was separated by SDS-PAGE and stained with silver, and the RBRP-GFP gel band was cut out and in-gel-digested with trypsin. For identification of endogenous RBRP peptide, the total proteins from tumor tissues were separated by tricine-SDS-PAGE and stained with silver. The gel bands with about 8 kDa were cut out and in-gel-digested with trypsin.

Protein identification was performed by MS as previously described[17]. MS assay was performed by Wininnovate Bio., Shenzhen. In brief, the extracted peptide mixtures, which were dissolved in buffer containing 0.1% formic acid and 2% acetonitrile (AcN), were eluted with a 5–40% gradient (0.1% formic acid and 95% AcN) and analyzed using nano-liquid chromatography–MS/MS AB SCIEX TripleTOF 5600, USA). Information-dependent acquisition mass spectrum techniques were used to acquire tandem MS data using a ion spray voltage of 2.4 kV, a air curtain pressure of 35 PSI, a atomizing pressure of 12 PSI, and an interface heating temperature of 150 °C. For an information dependent acquisition, survey scans were acquired in 250 ms and a maximum of 40 product ion scans (60 ms) were collected. Only the spectra with a charge state of two to four were selected to induce fragmentation by Collision-induced dissociation. A cycle time was set to 2.5 s. Dynamic exclusion was set to 16 s.

The WIFF RAW files were converted into peaklist files using Protein Pilot Software v4.5 (AB SCIEX). For the identification of RBRP interactors, protein identification was performed using the Mascot (v2.3.02) program against the Uniprot human protein database (released Dec 2014) with the default parameters. The false discovery rate was 1%, the protein score was ≥40, and unique peptides were ≥2.

For the identification of exogenous and endogenous RBRP peptide, protein identification was performed using the Mascot (version 2.6.2) program against our generated RBRP protein database with the default parameters. In brief, the following parameters were employed: the enzyme is trypsin; two missed cleavages were allowed; carbamidomethyl (C) was set as fixed modification, whereas acetyl (Protein N-term), deamidated (NQ), and oxidation (M) were considered as variable modifications. Precursor ion mass tolerance and fragment ion mass tolerance were 20 p.p.m. and 0.1 Da, respectively.

**RNA affinity purification**. Biotin-labeled RNA oligonucleotides containing A or m6A were synthesized by GeneScript (China). RNA affinity purification was performed as previously described[17]. Briefly, nuclear extracts were prepared from cells using a Nuclear and Cytoplasmic Protein Extraction Kit (Beyotime). Each 1 nmol of biotin-labeled RNA with A or m6A was bound to 100 μL of streptavidin–agarose beads (Sigma) overnight at 4 °C with rotation. RNA-immobilized beads were mixed with the nuclear mixture and incubated at 30 °C for 30 min. These beads were eluted by adding 30 μL of protein loading buffer and boiling for 5 min. The eluted mixtures were then analyzed using western blotting. These A- or m6A-oligo sequences were provided in Supplementary Table 3.

**RNA electrophoretic mobility shift assay**. RNA electrophoretic mobility shift assay (EMSA) was performed by LightShift RNA EMSA optimization and control kit (Thermo Fisher) following the manufacturer's instructions and a previous report[46]. Briefly, IGF2BP1-Flag proteins, WT and G19A-mutated RBRP oncopeptides, were overexpressed and purified from HEK293T cells according to the kit instructions (FLAGIPT1, Sigma) as previously described[41]. Biotin-labeled m6A-RNA oligos (20 pmol), 6 μg recombined IGF2BP-Flag proteins, together with 2 μg recombined RBRP-Flag oncopeptides or RBRP-Flag G19A mutant oncopeptides were mixed and incubated at room temperature for 30 min. Then, 1 μl of 0.2% glutaraldehyde was added to the mixtures and incubated on ice for 15 min. Then, the RNA-protein/oncopeptide mixtures were separated on 6% TBE (450 mM Tris, 450 mM boric acid, 10 mM EDTA pH 8.3) gel. The gel was transferred to a positively charged nylon transfer membrane (GE Healthcare) and the RNA membrane was crosslinked by UV light. The biotin-labeled m6A-RNA oligos were detected by a chemiluminescent nucleic acid detection module (Thermo Fisher) following the manufacturer's instructions. These A- or m6A-oligo sequences were provided in Supplementary Table 3.

**RIP assay**. RIP was performed as previously described with minor modifications[6]. In brief, cells were crosslinked by UV light and collected by trypsinization. Nuclear extracts were prepared from cells using a Nuclear and Cytoplasmic Protein Extraction Kit (Beyotime). Anti-IGF2BP1 or anti-HA antibodies, or a corresponding control IgG antibody were incubated with protein A/G agarose beads (Santa Cruz) for 4 h at 4 °C, followed by incubation with precleared nuclear extracts in RIP buffer (150 mM KCl, 25 mM Tris (pH 7.4), 5 mM EDTA, 0.5 mM dithiothreitol, 0.5% NP40, 1× protease inhibitor) supplemented with RNase inhibitor at 4 °C overnight. The co-IPed complexes were digested with DNase at 37 °C for 15 min and with proteinase K at 37 °C for 15 min. Input and co-IPed RNAs were extracted by TRIzol (Invitrogen) and determined by qPCR. The related enrichment of RNA in each sample was calculated by normalizing to input as previously described[6].

**Gene-specific m6A qPCR**. C-Myc mRNA m6A levels were determined as previously described with some modifications[6]. Briefly, total RNA was sheared by an

ultrasonicator (Covaris M220). An anti-m6A antibody or a corresponding control IgG antibody were incubated with protein A/G agarose beads (Santa Cruz) for 4 h at 4 °C, followed by incubation with the sheared RNAs in RIP buffer supplemented with RNase inhibitor for 6 h at 4 °C. The immunoprecipitated m6A-methylated RNAs were digested with DNase at 37 °C for 15 min and with proteinase K at 37 °C for 15 min, then extracted by TRIzol (Invitrogen) and determined by qPCR. The related enrichment of m6A in each sample was calculated by normalizing to input.

**RNA m6A dot blotting**. RNA dot blotting was performed as previously described with minor modifications[41]. Briefly, total RNAs or RIP-immunoprecipitated RNAs were placed at 55 °C for 15 min to denature in a denaturing buffer (2.2 M formaldehyde, 50% deionized formamide, 0.5× MOPS buffer). RNA was heated at 95 °C for 10 min and cooled on ice for 5 min. RNAs were then spotted onto Amersham Hybond-N+ membranes (GE Healthcare, cat# RPN303B) and cross-linked by UV light. The m6A modifications on RNA were detected with an anti-m6A antibody.

**Half-life of mRNA and mRNA stability assay**. Cells at 70–80% confluency were treated with 5 μg/ml actinomycin D and collected at the indicated time points. Total RNA was extracted by TRIzol (Invitrogen) and the *c-Myc* mRNA levels were determined by RT-PCR. The intensity of gel band was measured by ImageJ (v1.8.0) program. The turnover rate and half-life of mRNA were calculated according to previously reported methods[6].

**Construction of synonymous mutants**. Synonymous mutants of *IGF2BP1-HA* and *c-Myc-HA*, which are resistant to the anti-*IGF2BP1* and *c-Myc* siRNAs, respectively, were constructed. The anti-*IGF2BP1* siRNA-targeted sequence 5′-CTCCGGGAAAGTAGAATT-3′ in WT and GxxGΔ-mutated *IGF2BP1-HA* plasmids was synonymously mutated to 5′-CTCAGGAAAGGTGGAGTT-3′. The synonymous mutants s*IGF2BP1-HA* and s*IGF2BP1-HA* GxxGΔ were generated. The anti-*c-Myc* siRNA-targeted sequence 5′-GTGCAGCCGTATTTCTACT-3′ in WT and CRDmut-mutated *c-Myc-HA* plasmids was synonymously mutated to 5′-GTACAACCATACTTTTATT-3′. The synonymous mutants s*c-Myc-HA* and s*c-Myc-HA* CRDmut were generated.

**Statistical analyses and reproducibility**. Two-tailed unpaired Student's *t*-test without adjustment was used to compare data between two groups, unless explicitly stated. Mann–Whitney *U*-test was used to compare clinical data between two groups and has been stated in figure legends. The significance of the growth curves was analyzed using a two-way analysis of variance. The treatment groups were compared with the control, unless stated otherwise. Survival curves were analyzed using the Kaplan–Meier method and compared using the log-rank test. Statistical analyses were performed using Prism 5 or Prism 8 software and the SPSS program. $n = 3$ independent experiments unless stated. Two independent experiments were performed for immunofluorescence staining assays. One independent experiment was performed for the remaining results. The data are presented as means ± SD, unless indicated otherwise; $*p < 0.05$, $**p < 0.01$, $***p < 0.001$ and ns, no statistical significance.

**Reporting summary**. Further information on research design is available in the Nature Research Reporting Summary linked to this article.

## Data availability

The mass spectrometry proteomics data for identification of exogenous and endogenous RBRP peptide have been deposited to the ProteomeXchange Consortium (http://proteomecentral.proteomexchange.org) via the iProX[47] partner repository with the dataset identifiers PXD016267 and PXD016268. The larger figures for all of the cytology images including the immunofluorescence images (Figs. 1a, b, 2a, 3c, d, 7d, e, and Supplementary Figs. 2a, b, e, 5b, c, e, f, 6c, d, 7c, d, 11c, d) and the uncropped gels and blots (Figs. 1c, 2b, c, 3a, 4a–j, 5a, c, d, f, g, 6c, g, i and Supplementary Figs. 2c, d, 4a, c, 6a, 7a, 8a–d, 9b, c, 10a–e, 11a) are provided as a Source file. The raw data for all of our histograms and line charts (Figs. 2e–g, 3b–e, g, 5b, e, h, i, 6a, b, d–f, h, j, k, 7a–e and Supplementary Figs. 2c, 4b, d, 5a–f, 6b–d, 7b–d, 9a, d, 11b–d) are provided as a Source Data file. All other relevant data are available from the corresponding author on reasonable request.

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

## Acknowledgements

This work was supported by the National Natural Science Foundation of China (81772998, 81672393, and 81802763), the R&D Plan of Guangzhou (201704020115), the Yangcheng Scholars program from the Ministry of Education of Guangzhou (1201561583), Innovative Research Team of Ministry of Education of Guangzhou (1201610015), the R&D Plan of Guangdong (2017A020215096).

## Author contributions

G.-R.Y. conceived the project, designed the experiments, and wrote the paper. S.Z., J.-Z.W., Y.-T.H., N.M., M.C., R.-X.L., X.-H.C., and X.-L.Z. conducted the in vitro and in vivo experiments. D.C. collected the clinical samples. S.Z. and J.-Z.W. provided the statistical analyses.

## Competing interests

The authors declare no competing interests.
