## [Peer Review File · Nature Communications]

Reviewers' comments:

Reviewer #1 (Remarks to the Author): Expert in m6A

In this manuscript, Zhu S. et al. reported a very interesting and novel discovery that a small peptide (namely RNA-binding regulatory peptide; RBRP) encoded by a lncRNA (LINC00266-1) plays an oncogenic role in colorectal cancer (CRC) by directly binding to IGF2BP1 and enhancing the recognition of IGF2BP1 on N6-methyladenosine (m6A)-modified RNA transcripts (e.g., c-Myc). First, the authors demonstrated that LINC00266-1 encodes a 71 aa novel peptide (i.e., RBRP) in both forced expression system and endogenous cellular system. They showed that expression of RBRP was increased in highly metastatic colorectal, ovarian, nasopharyngeal and breast cancer cell sublines, compared to the levels in their parental cell lines, and that its expression was also increased in the primary CRC tissues relative to that in the corresponding nontumoral colorectal tissues. Moreover, they showed that high level of RBRP peptide is an independent predictor for the poor prognosis in patients with CRC. Next, they conducted both in vitro and in vivo functional studies and demonstrated that RBRP peptide, but not the host lncRNA, promotes tumorigenesis. Mechanistically, they showed that RBRP interacts directly with IGF2BP1, an RNA m6A reader protein that was reported recently to be able to promote the stability and translation of m6A-modified target transcripts (Huang H, et al. NCB. 2018). More specifically, they demonstrated that G19 of RBRP and the KH3-4 di-domain of IGF2BP1 are the most essential residue or domain for their direct interaction, which at least in part explained why the KH3-4 di-domain was functionally essential for IGF2BP proteins as m6A readers (Huang H, et al. NCB. 2018). Then, they confirmed that RBRP promotes tumorigenesis through binding with IGF2BP1 and strengthening the m6A recognition by IGF2BP1 on target RNA transcripts (e.g., c-Myc). They further showed that RBRP promotes the stability of c-Myc mRNA by enhancing the function of IGF2BP1, and that c-MYC is a functionally essential target of the RBRP/IGF2BP1 axis in tumorigenesis. Overall, this is an interesting, timely, comprehensive and significant study, which shows for the first time that a novel peptide, encoded by a previously functionally unknown lncRNA, binds to a m6A reader protein (herein IGF2BP1) and enhances the reader protein's capacity in recognition of m6A-modified RNA transcripts and in promoting the stability of the target transcripts (e.g., C-Myc), leading to enhanced cancer cell growth/proliferation, migration, invasion and tumorigenesis. This study is of high novelty and significance, which reveals a previously unappreciated connection between a lncRNA, a small peptide encoded by the lncRNA, and the m6A regulatory pathway, and provide novel insights into the mechanisms underlying m6A recognition and tumorigenesis.

I have only a few minor concerns for the authors to address:

- (1) The authors should list the detailed sequence information for the following sequences: LINC00266-1 ORF-Flag, 5'UTR-ORF-Flag, and 5'UTR-ORFmut-Flag, as well as c-Myc CRD1 and CRD2 mRNA oligos with A or m6A (see Figure 5), and m6A-unmethylated (ss-A) or methylated (ss-m6A) single-strand RNA probes (see Figure s8).
- (2) The multiple lines shown in Figure 3b are clustered together and thus it is very difficult to distinguish them. The authors should find a better way to illustrate them.
- (3) On Page 10, "In these cells, we restored the expression of LINC00266-1 ORF-Flag or 5'UTR-ORFmut-Flag, which..." should be "In these cells, we restored the expression of LINC00266-1 ORF-Flag (ORF), 5'UTR-ORF-Flag (5'U) or 5'UTR-ORFmut-Flag (MT), which..."

Reviewer #2 (Remarks to the Author): Expert in lncRNA

In their manuscript entitled "An oncopeptide regulates m6A recognition by the m6A reader IGF2BP1 and tumorigenesis", the authors discover a novel peptide derived from a formerly annotated lncRNA which binds to IGF2BP1 and affects its interaction with methylated RNAs like c-MYC as well as the RNA stabilizing proteins HuR, MATR3 and PABPC1. Overall, the study provides a highly comprehensive and cosmetic dataset including many controls and a wealth of data. The manuscript is well-written and well-structured. Prior to publication, I

mainly have a number of questions or suggestions for improvements.

1) Knockdown experiments

Many of the knockdown experiments have been carried out with only a single siRNA or shRNA, which makes it prone to off-target effects. In several instances, this is remedied by a rescue by the overexpression of the target gene. In other instances, it may be justified by the complexity of the experiment by combining the knockdown with the overexpression of multiple different plasmids. Nonetheless, for critical experiments, at least a second, independent RNAi reagent would be desirable - e.g. to show the rescue of RBRP overexpression phenotypes by knockdown of c-MYC. Notably, the authors also write in the text and legend about "anti-c-MYC siRNAs", but in fact, only one siRNA seems to have been used.

2) Quantification

The authors should double-check whether the quantification of their data accurately reflects the actual results. For example, in supplementary figure S11c, I cannot see the differences depicted in the quantification in the primary data next to them, especially for HCT-116.

3) Interactome

The authors seem to have identified quite a number of different proteins interacting with RBRP (bands figure 4a). The complete set of interaction partners is provided in supplementary table 4 (should be referred to in the manuscript). The possible role of other interactors should be discussed, the rationale for selecting IGF2BP1 should be explained and the specificity of the interaction given that more than a hundred candidates were identified.

4) Literature

The authors should discuss the state-of-the-art on lncRNA-derived peptides in cancer (e.g. PMID 29765154, PMID 30367041) and compare their study to these previous studies.

5) Methods

The methods should be described in more detail. For example, I could not find a table of all the siRNA and shRNA sequences used as well as how these were transfected (e.g. concentration).

Dear Reviewers,

Thank you very much for commenting our manuscript and giving us the opportunity to revise our manuscript entitled “An oncopeptide regulates m⁶A recognition by the m⁶A reader IGF2BP1 and tumorigenesis”. Your constructive comments and suggestions are helpful and greatly appreciated. Accordingly, we have made the suggested revisions in the manuscript and provided a point-by-point response to these comments as follows.

We hope that you will find this revision satisfactory and our revised manuscript acceptable for publication in Nature Communications.

Sincerely,

Guang-Rong Yan, PhD

Professor

The Third Affiliated Hospital

Guangzhou Medical University

Referee: 1

Question #1:

The authors should list the detailed sequence information for the following sequences: LINC00266-1 ORF-Flag, 5'UTR-ORF-Flag, and 5'UTR-ORFmut-Flag, as well as c-Myc CRD1 and CRD2 mRNA oligos with A or m6A (see Figure 5), and m6A-unmethylated (ss-A) or methylated (ss-m6A) single-strand RNA probes (see Figure s8).

Response:

We have added the detailed sequence information for your indicated sequences in the supplementary Table s5. Please find and check them.

Question #2:

The multiple lines shown in Figure 3b are clustered together and thus it is very difficult to distinguish them. The authors should find a better way to illustrate them.

Response:

I have distinguished these lines by zooming in on the image and modifying the line color. And the raw data of Figure 3b have provided in the "Source Data" file which will be online published with our manuscript. The readers can see the larger Figure 3b in which multiple lines were distinguished.

Question #3:

On Page 10, "In these cells, we restored the expression of LINC00266-1 ORF-Flag or 5'UTR-ORFmut-Flag, which..." should be "In these cells, we restored the expression of LINC00266-1 ORF-Flag (ORF), 5'UTR-ORF-Flag (5'U) or 5'UTR-ORFmut-Flag (MT), which..."

Response:

Thanks. We have revised it.

Referee: 2

Question #1:

Many of the knockdown experiments have been carried out with only a single siRNA or shRNA, which makes it prone to off-target effects. In several instances, this is remedied by a rescue by the overexpression of the target gene. In other instances, it may be justified by the complexity of the experiment by combining the knockdown with the overexpression of multiple different plasmids. Nonetheless, for critical experiments, at least a second, independent RNAi reagent would be desirable - e.g. to show the rescue of RBRP overexpression phenotypes by knockdown of c-MYC. Notably, the authors also write in the text and legend about "anti-c-MYC siRNAs", but in fact, only one siRNA seems to have been used.

Response:

We have added and performed the related experiments using anti-c-Myc siRNA#2 and have added the related results in supplementary Figure s11. As same as the results using anti-c-Myc siRNA#1, the increases in cell proliferation, colony formation, and migration and invasion induced by RBRP overexpression were completely blocked by *c-Myc* knockdown using anti-c-Myc siRNA#2. The two anti-c-Myc siRNA sequences have been provided in the supplementary Table s5.

Question #2:

The authors should double-check whether the quantification of their data accurately reflects the actual results. For example, in supplementary figure S11c, I cannot see the differences depicted in the quantification in the primary data next to them, especially for HCT-116.

Response:

Sorry, as shown below, we did not choose the most appropriate representative image in Figure s11c for HCT-116.

ORF-Flag	NC	NC	ORF	ORF	MT	MT
Sic-Myc	NC	+	NC	+	NC	+

We re-performed the colony assay and the corresponding quantification analyses. The quantification of their data accurately reflects the actual results in the revised supplementary Figure s11c. And the related raw data have been also provided in the “Source Data” file.

Question #3:

The authors seem to have identified quite a number of different proteins interacting with RBRP (bands figure 4a). The complete set of interaction partners is provided in supplementary table 4 (should be referred to in the manuscript. The possible role of other interactors should be discussed, the rationale for selecting IGF2BP1 should be explained and the specificity of the interaction given that more than a hundred candidates were identified.

Response:

Sorry, we forgot to refer to the supplementary Table s4 in the manuscript. We have added it.

It is a good suggestion. We have added the rationale for selecting IGF2BP1 in the “Results” section and have discussed the possible role of other interactors in the “Discussion” section.

107 potential interactors with RBRP were here identified by interactomics assay. A

gene annotation assay showed that 76 of these 107 proteins belong to the RNA-binding proteins. The RNA-binding protein, RNA m6A reader IGF2BP1 was particularly interesting because the protein score of IGF2BP1 was TOP5 in the RBRP-interactomics identification; RNA m6A modification play an important role in tumorigenesis and cancer progression; and IGF2BP1 regulates the level of one of the most important oncogene c-Myc and tumorigenesis, consistent with the oncogenic roles of RBRP in tumorigenesis. Therefore, we select IGF2BP1 for further investigation in this study. In the manuscript, we have identified the interaction of RBRP with RNA-binding protein, RNA m6A reader IGF2BP1 and elucidated its mechanism.

It is likely that other interactors with RBRP may also contribute to the hallmark of cancer. Among these RBRP-interacting proteins, the gene annotation assay demonstrated that these RBRP-interactors mainly involve in the regulation of RNA splicing, and many splicing regulators interact with RBRP including 10 heterogeneous ribonucleoprotein (hnRNP) family proteins, 7 splicing factors and 2 small nuclear ribonucleoproteins (snRNPs), suggesting that RBRP may regulate pre-mRNA splicing.

[REDACTED]

Here, we have discussed the possible role of other interactors in the “Discussion” section according to the reviewer’s suggestion.

[REDACTED]

Question #4:

The authors should discuss the state-of-the-art on lncRNA-derived peptides in cancer (e.g. PMID 29765154, PMID 30367041) and compare their study to these previous studies.

Response:

We have discussed these previous studies in the “Discussion” section.

Question #5:

The methods should be described in more detail. For example, I could not find a table of all the siRNA and shRNA sequences used as well as how these were transfected (e.g. concentration).

Response:

We added the “RNA interference” section in the supplementary Methods section. The transfected conditions (e.g. concentration) were addressed. And we have added the detailed sequence information for all the siRNA, shRNA, c-Myc CRD1 and CRD2 mRNA oligos with A or m6A, and m6A-unmethylated (ss-A) or methylated (ss-m6A) single-strand RNA probes in the supplementary Table s5. Please find and check them.

REVIEWERS' COMMENTS:

Reviewer #1 (Remarks to the Author):

In this revised manuscript, the authors have conducted a set of new experiments and added additional data, and have carefully and thoroughly addressed all the critiques from the reviewers. As a result, the quality of this paper has been further improved. Overall, this is an interesting and timely study. Therefore, I fully support the publication of this paper in Nature Communications.

Reviewer #2 (Remarks to the Author):

The authors had adequately addressed my previous concerns.

REVIEWERS' COMMENTS:

Reviewer #1 (Remarks to the Author):

In this revised manuscript, the authors have conducted a set of new experiments and added additional data, and have carefully and thoroughly addressed all the critiques from the reviewers. As a result, the quality of this paper has been further improved. Overall, this is an interesting and timely study. Therefore, I fully support the publication of this paper in Nature Communications.

Reviewer #2 (Remarks to the Author):

The authors had adequately addressed my previous concerns.

Response: Thank your review and kind help very much.